# Local integrals of motion
# in quasiperiodic many-body localized systems

Steven J. Thomson[1,2*] and Marco Schirò[2]

**1** Dahlem Center for Complex Quantum Systems, Freie Universität, 14195 Berlin, Germany
**2** JEIP, UAR 3573 CNRS, Collège de France, PSL Research University,
11 Place Marcelin Berthelot, 75321 Paris Cedex 05, France

★ steven.thomson@fu-berlin.de

## Abstract

Local integrals of motion play a central role in the understanding of many-body localization in many-body quantum systems in one dimension subject to a random external potential, but the question of how these local integrals of motion change in a deterministic quasiperiodic potential is one that has received significantly less attention. Here we develop a powerful new implementation of the continuous unitary transform formalism and use this method to directly compute both the effective Hamiltonian and the local integrals of motion for many-body quantum systems subject to a quasiperiodic potential. We show that the effective interactions between local integrals of motion retain a strong fingerprint of the underlying quasiperiodic potential, exhibiting sharp features at distances associated with the incommensurate wavelength used to generate the potential. Furthermore, the local integrals of motion themselves may be expressed in terms of an operator expansion which allows us to estimate the critical strength of quasiperiodic potential required to lead to a localization/delocalization transition, by means of a finite size scaling analysis.



# 1   Introduction

The study of quenched random disorder in non-interacting quantum systems dates back to the seminal work of Anderson [1] who first understood how diffusion could eventually break down in the presence of strong enough randomness, leading to localization of the single-particle wavefunctions. It was later realized [2] that this localization transition occurs in three dimensional systems, while lower dimensional non-interacting quantum systems are always localized for any small disorder. The complex interplay of quantum many-body interactions and *random* disorder was studied thereafter [3], but has been the subject of renewed attention over the last decade since the theoretical prediction of localization at finite temperatures in interacting systems [4], followed by rapid experimental verification [5–7] and further theoretical progress, summarized in Refs. [8–11]. Now known as many-body localization (MBL), this is primarily a dynamical effect whereby highly excited eigenstates may spontaneously fail to thermalize due to disorder, and is typically understood in terms of the existence of an extensive number of locally conserved quantities known as integrals of motion [12–15] (LIOMs, or $l$-bits) which prevent the system from thermalizing, in contrast to ergodic systems which conserve only global quantities such as the total energy or number of particles. The breakdown of MBL is understood in terms of resonances, i.e. spatially separated sites with on-site energies which differ by less than the hopping amplitude, facilitating transport in the system. Proliferation of these resonances can lead to so-called avalanche effects, which can lead to transport throughout the entire system and consequently destroy localization [16, 17]. Understanding the role of these resonances and their distribution within a given system is crucial in order to understand the stability of disorder-induced localization.

While many-body localization in systems subject to a random external potential is by now largely understood in terms of LIOMs, in some cases rigorously [18], there is no corresponding rigorous understanding of MBL phenomena in systems with quasiperiodic potentials of the sort

routinely realised in current experiments, despite the fact that both experimental [19–22] and numerical evidence exists for MBL phenomena in such systems [23–27]. Quasiperiodic potentials may be loosely understood as intermediate between periodic and random potentials, and are characterised by a set of discrete but non-periodic Fourier components. First studied contemporaneously with Anderson localization [28] and further investigated in the decades since [29], they exhibit many crucial differences with respect to random disorder as the potential is strongly correlated in real space. This dramatically changes the spatial structure of resonances already in single-particle models [30], enhancing the probability of encountering resonances between spatially separated sites, and could have important implications for the stability of many-body localized phases of matter in quasicrystalline materials [31]. Furthermore rare-region Griffiths effects [32], which are responsible for much of the exotic phenomenology of MBL systems on both sides of the transition and possibly also for the critical behavior [33–37], are absent in quasiperiodic systems. This points towards qualitatively different physics at play in the quasiperiodic MBL problem, as well as towards a different mechanism responsible for its eventual instability towards full delocalisation, an issue which is still far from being fully understood [38]. Recently there have been several studies presenting contrasting results for the nature of the MBL transition in quasiperiodic systems, with the level statistics obtained from exact diagonalization [27] giving a rather different result for the transition point and critical exponent than renormalization group calculations [39] and studies based around the construction of LIOMs using time-averaged local observables [40], as first proposed in Ref. [41].

In this work we explicitly construct the LIOMs for a model of interacting one dimensional fermions in a quasiperiodic potential and use them to gain insight on the nature of the MBL phase and its possible transition towards delocalization. To this extent we develop a new computational implementation of the established flow equation method, leveraging the high degree of parallelization possible with modern computer hardware and tensor algebra operations. We show that this *tensor flow equation* approach leads to a significant improvement upon previous implementations [42,43] in both accuracy, thanks to the inclusion of additional running couplings in the flow parametrization, and transparency. Using this new method we compute explicitly the LIOM interactions and real-space support, showing how they contain clear fingerprints of the quasiperiodic structure of the potential in the form of local dips or peaks at distances given by the associated asymptotic numbers. Furthermore we show that the richer structure of the tensor flow equation and of its LIOM description allows us to build up a diagnostic for the MBL phase transition, so far out of reach for previous implementations of the method, that we estimate here for the quasiperiodic problem using finite-size scaling. We end by discussing our results in the contexts of other recent related works [39, 40, 44].

We wish to emphasise here at the outset that by comparison with previous works making use of truncated flow equation methods [42, 43, 45], this new work offers the following major advantages:

- Inclusion of high-order off-diagonal terms in the ansatz for the running Hamiltonian, which leads to a significant improvement in the accuracy and stability of the method.

- Ability to compute the many-body contributions to the LIOM operator expansion, which in turn allows us to construct a diagnostic for the MBL phase transition.

- Formulation in terms of tensors and tensor contractions allows us to make use of highly efficient numerical linear algebra routines in order to implement the technique in an entirely new way that is not only faster to compute, but is more transparent and offers a clear path towards systematic extension of the method to higher orders and other systems of interest.

## 2 Model

We start from a one-dimensional fermionic Hamiltonian of the form:

$$\mathcal{H} = \sum_i h_i : n_i : + J \sum_i (: c_i^\dagger c_{i+1} : + H.c.) + \Delta_0 \sum_i : n_i n_{i+1} :, \tag{1}$$

where the on-site terms $h_i$ are drawn from a quasiperiodic potential given by $h_i = W \cos(2\pi i/\phi + \theta)$ where $\phi$ is some irrational number, $\theta$ is a random phase and $W$ is the amplitude of the quasiperiodic potential (i.e. plays the role of 'disorder strength'), with nearest-neighbour hopping $J$ and interactions $\Delta_0$. For convenience, we shall refer to the choice of phase $\theta$ as a 'disorder realization' and to averages over different values of $\theta$ as 'disorder averaging', though it should be noted that quasiperiodic potentials are not true disorder as they are entirely deterministic and there is no randomness. The notation $: \cdots :$ signifies vacuum normal-ordering, which is necessary to enforce a consistent ordering of operators - see Appendix A for details.

In the absence of interactions, i.e. $\Delta_0 = 0$, Eq. 1 reduces to the well known Aubry-André (or Aubry-André-Harper) model [28, 29, 46], which despite being a single-particle system exhibits a great deal of non-trivial physics. This model undergoes a localization transition at a quasiperiodic potential strength of $W_c/J = 2$, which can be shown analytically through the self-duality of the Hamiltonian. This is unlike non-interacting one-dimensional systems subject to a random external potential, which are always Anderson localized and do not exhibit a delocalization transition. Above this critical value of $W_c/J$, the single-particle wavefunctions decay exponentially in space, but with quasiperiodic revivals due to the structure of the underlying potential. Precisely at the critical point, the single particle spectrum is fractal in nature, and takes the form of a Cantor set [47]. The Aubry-André model does not have a mobility edge, and although variations exist which do exhibit single-particle mobility edges [48, 49], we will not consider those modifications in the present work.

The interacting Aubry-André model is comparatively less understood, although it has received some attention in recent years motivated by the MBL problem [23–27, 31, 39]. In the limit of weak interactions, the bosonic Aubry-André model in one dimension has been shown to exhibit an inverted mobility edge, with the high-energy excitations appearing exponentially localized while the low-energy excitations remain extended [50]. The low-energy properties of bosonic versions of the interacting Aubry-André model have been studied recently in both one [51] and two [52, 53] spatial dimensions and shown to exhibit unusual fractal properties. Interacting quasiperiodic spin chains in one dimension have been studied using tensor network techniques [54] to simulate their non-equilibrium dynamics, and recently with exact diagonalization [40] in order to construct time-averaged local integrals of motion. Fermionic Aubry-André models have also been studied in the context of many-body localization using quantum quench spectroscopy to identify mobility edges [55]. Here we will focus on the one-dimensional fermionic model (which maps onto the spin model via a Jordan-Wigner transform), however the method we outline in the following section is not unique to fermions and may also be applied to interacting bosons with only minor modifications.

## 3 Method

Here we will first review the use of continuous unitary transforms, also known as flow equation methods, before detailing the new implementation we use in the present manuscript which we find offers many practical advantages over previous implementations, while still sharing the same underlying philosophy. The new implementation makes use of the Python package

PyFlow [56], developed by one of the present authors, and running on graphics processing units (GPUs).

## 3.1 Brief Review of Continuous Unitary Transforms

The use of continuous unitary transforms to diagonalize Hamiltonians has a long and varied history. The technique was originally proposed in condensed matter physics by Wegner [57] under the name 'flow equations' (later popularized by Kehrein [58] and coworkers [59–63]), independently in high-energy physics by Glazek and Wilson under the name 'similarity transform' [64,65], and also in mathematics under the names 'isospectral flow' and 'double bracket flow' [66–68]. Since then the method has also been generalized to time-dependent systems in a variety of forms [59,69–72] including driven and dissipative scenarios, however here we will focus on Hamiltonians with no explicit time dependence.

The core of the method is the idea of using a series of infinitesimal unitary transformations to diagonalise a particular system of interest. This is spiritually similar to the well-known Schrieffer-Wolff [73,74] transform, where a Hamiltonian may be diagonalized to leading order by careful choice of a suitable unitary transformation:

$$\tilde{\mathcal{H}} = e^S \mathcal{H} e^{-S} = \mathcal{H} - [\mathcal{H}, S] + \dots, \tag{2}$$

and choosing the generator $S$ such that $[\mathcal{H}, S]$ is equal to the off-diagonal terms of the Hamiltonian which we wish to remove. In general, this procedure will also generate new higher-order terms which must then be removed by further transforms, or treated in some other perturbative manner. Here, rather than making a single 'large' unitary transform, we instead make a series of infinitesimal transforms, each of which can be made arbitrarily accurate:

$$\mathcal{H}(l + \mathrm{d}l) = e^{\eta(l)\mathrm{d}l} \mathcal{H}(l) e^{-\eta(l)\mathrm{d}l} = \mathcal{H}(l) + \mathrm{d}l \, [\eta(l), \mathcal{H}(l)], \tag{3}$$

where $l$ is a fictitious 'flow time' which parameterises the transform, and $\eta(l)$ is some scale-dependent anti-Hermitian generator chosen to diagonalize the specific problem of interest in the $l \to \infty$ limit. In the spirit of renormalization group techniques, the transformation of the Hamiltonian can be written in terms of a single so-called flow equation:

$$\frac{\mathrm{d}\mathcal{H}(l)}{\mathrm{d}l} = [\eta(l), \mathcal{H}(l)], \tag{4}$$

such that the eventual 'fixed point' of the flow is a diagonal Hamiltonian, with $[\eta(l \to \infty), \mathcal{H}(l \to \infty)] = 0$. The properties of the final fixed-point Hamiltonian are controlled by the generator $\eta(l)$, which we can freely choose to be any anti-Hermitian operator such that the overall transform is unitary. The choice of generator is far from unique and various options exist in the literature which result in a diagonal fixed-point Hamiltonian [75,76], however a common choice is the so-called 'Wegner generator':

$$\eta(l) = [\mathcal{H}_0(l), V(l)], \tag{5}$$

where $\mathcal{H}_0$ represents the diagonal terms in the Hamiltonian and $V = \mathcal{H} - \mathcal{H}_0$ contains the off-diagonal terms which we want to vanish in the $l \to \infty$ limit. The Wegner generator is suitable for most problems as it a robust choice that can be stably numerically integrated, although it has the significant disadvantage for sparse models (i.e. those with nearest-neighbour couplings only) that it does not preserve the sparsity of the initial Hamiltonian, and in the early stages of the flow it typically generates long-range off-diagonal couplings which must be kept track of until they later decay to zero. Other choices of generator exist which do preserve the sparsity of nearest-neighbour models [75], however they are typically much less numerically stable [77],

and so we do not consider them here. One interesting frontier is the development of adaptive generators, following an intriguing proposal for adaptive Schrieffer-Wolff transforms [78], however this is still an active area of development and in the present work we focus on the established Wegner generator.

Flow equations in disordered systems have been the subject of increasing attention over the last few years with a variety of implementations including non-interacting systems with long-range couplings [77,79], numerically exact studies of interacting systems [80,81], analytical treatments proposing efficient generators [75] and approximate methods for interacting systems [42, 43, 45] which made use of a truncation in operator space to limit the number of running couplings to a computationally tractable number even for large systems, enabling studies of MBL phenomenology not only in one-dimension, but also in coupled chains and even in two spatial dimensions.

Here we will build on the latter approach, employing a truncation in operator space together with an entirely new implementation more suited to the efficient linear algebra capabilities of modern computer hardware and extending the truncation to include new higher-order terms which, as we shall show, dramatically increases the accuracy of the method and enables us to do away with some of the complexities required in earlier works. Moreover, this new implementation allows us to compute new quantities which were out of the reach of previous implementations, including the identification of a delocalization transition in a quasiperiodic many-body system.

## 3.2 Tensor Flow Equations

To illustrate the new implementation, which we call the Tensor Flow Equation (TFE) technique, here we start from a very general normal-ordered (fermionic) Hamiltonian with quadratic and quartic terms:

$$\mathcal{H} = \sum_{ij} \mathcal{H}_{ij}^{(2)} + \sum_{kqlm} \mathcal{H}_{kqlm}^{(4)} \tag{6}$$

$$= \sum_{ij} H_{ij}^{(2)} : c_i^\dagger c_j : + \sum_{kqlm} H_{kqlm}^{(4)} : c_k^\dagger c_q c_l^\dagger c_m : . \tag{7}$$

For simplicity we will assume this Hamiltonian to describe a one-dimensional chain of length $L$, however this procedure can easily be generalized to two-dimensional models, as with previous implementations of flow equations [42]. In all of the following, the bracketed superscripts indicate how many fermionic operators are associated with a term. The couplings in this Hamiltonian may be short- or long-ranged, and may be homogeneous or disordered, therefore this form incorporates a large range of systems of interest.[1] This Hamiltonian can be represented schematically with the following diagram:

$$\mathcal{H} = \boxed{\mathcal{H}^{(2)}} + \boxed{\mathcal{H}^{(4)}}$$

where $\mathcal{H}^{(2)}$ is a rank-2 tensor representing the quadratic part of the Hamiltonian, $\mathcal{H}^{(4)}$ is a rank-4 tensor representing the quartic part of the Hamiltonian, the 'out' arrows signify fermionic creation operators ($c_i^\dagger$), and the 'in' arrows signify fermionic annihilation operators ($c_j$). By

---

[1]While the Hamiltonian used here is very general, one must however take care that the generator of the unitary transform is non-zero: for noninteracting translationally invariant systems, for example, the Wegner generator which we use in this manuscript vanishes and the unitary transform we employ here reduces to the identity. In such cases, one must use a different generator or work in momentum space rather than real space.

construction, the end result of our procedure will be a Hamiltonian which is diagonal in terms of the fermionic number operators and will take the form:

$$\tilde{\mathcal{H}} = \sum_i \tilde{H}_{ii}^{(2)} \tilde{n}_i + \sum_{ij} \tilde{H}_{iijj}^{(4)} \tilde{n}_i \tilde{n}_j + \dots,\qquad(8)$$

$$\tilde{\mathcal{H}} = \boxed{\tilde{\mathcal{H}}^{(2)}} + \boxed{\tilde{\mathcal{H}}^{(4)}}$$

where the tilde notation indicates that all quantities are given in the diagonal basis and the ellipsis indicates possible higher-order terms which we shall discuss later. For flow equations, the key ingredient is the calculation of commutation relations between different parts of the Hamiltonian. For example, the canonical Wegner generator is given by:

$$\eta = [\mathcal{H}_0, V] = \left[\mathcal{H}^{(2)}, V^{(2)}\right] + \left[\mathcal{H}^{(4)}, V^{(2)}\right] + \left[\mathcal{H}^{(2)}, V^{(4)}\right] + \dots,\qquad(9)$$

where $V$ represents the off-diagonal part of the Hamiltonian, which we can choose freely.

Let's take these three commutators individually:

$$i)\left[\mathcal{H}^{(2)}, V^{(2)}\right] = \sum_{ijk}\left(\mathcal{H}_{ik}^{(2)} V_{kj}^{(2)} - V_{ik}^{(2)} \mathcal{H}_{kj}^{(2)}\right),\qquad(10)$$

which is just a series of matrix contractions which can be represented schematically by

$$\boxed{\mathcal{H}^{(2)}}\quad \boxed{V^{(2)}} \quad - \quad \boxed{V^{(2)}}\quad \boxed{\mathcal{H}^{(2)}}$$

where the line joining the two legs marked $k$ represents a sum over this index, i.e. a matrix/tensor contraction. These tensor contractions can be efficiently computed numerically by most standard linear algebra packages, and can be effectively parallelised over multiple cores - if the arrays fit in memory, then graphical processing units (GPUs) are ideal for this, as modern GPUs typically have $10^3 - 10^4$ cores, compared to even a cluster computer which may have only $\sim 10^2$ cores per node, and this can result in a significant speed increase - see Appendix C for benchmarks demonstrating this.

We can follow the same procedure for the higher order terms, where the commutator (see Appendix A) can be written as:

$$ii)\left[\mathcal{H}^{(4)}, V^{(2)}\right] = \sum_{ijkql}\left(\mathcal{H}_{ljkq}^{(4)} V_{il}^{(2)} + \mathcal{H}_{ijlq}^{(4)} V_{kl}^{(2)} + \mathcal{H}_{ilkq}^{(4)} V_{lj}^{(2)} + \mathcal{H}_{ijkl}^{(4)} V_{lq}^{(2)}\right).\qquad(11)$$

This can be represented graphically as the sum of all possible two-point tensor contractions where each index is from a different tensor:

$$\boxed{\mathcal{H}^{(4)}}\ \boxed{V^{(2)}} + \boxed{\mathcal{H}^{(4)}}\ \boxed{V^{(2)}} + \boxed{\mathcal{H}^{(4)}}\ \boxed{V^{(2)}} + \boxed{\mathcal{H}^{(4)}}\ \boxed{V^{(2)}}$$

where the contracted indices are summed over. This reproduces the same results as obtained in Appendix A using a lengthier algebraic procedure. After evaluating the contractions, this can be written graphically as:

$$\mathcal{H}^{(4)}_{ljkq} V^{(2)}_{il} \quad + \quad \mathcal{H}^{(4)}_{ijlq} V^{(2)}_{kl} \quad + \quad \mathcal{H}^{(4)}_{ilkq} V^{(2)}_{lj} \quad + \quad \mathcal{H}^{(4)}_{ijkl} V^{(2)}_{lq}$$

$$\qquad\qquad\quad j\ k\ q\ i \qquad\qquad\quad i\ j\ q\ k \qquad\qquad\quad i\ k\ q\ j \qquad\qquad\quad i\ j\ k\ q$$

where here we use the Einstein summation convention such that any repeated indices are summed over. By swapping the order of indices and keeping in mind fermionic anti-commutation relations for normal-ordered operators, we can collect together the terms with the correct $\pm$ prefactors.

$$-\mathcal{H}^{(4)}_{ljkq} V^{(2)}_{il} - \mathcal{H}^{(4)}_{ijlq} V^{(2)}_{kl} + \mathcal{H}^{(4)}_{ilkq} V^{(2)}_{lj} + \mathcal{H}^{(4)}_{ijkl} V^{(2)}_{lq}$$

$$\qquad\qquad\qquad i \qquad\quad j \qquad\qquad k \qquad\quad q$$

Here we can start to see why the tensor notation is useful - the contractions and the signs can be read off from the graphical notation. This also enables us to generalise the above method easily to bosons, without having to work out Wick's Theorem explicitly in full to compute the products of normal-ordered operators (see Appendix A), as the commutation relations are automatically taken into account when re-ordering the indices. By the same token, this graphical notation allows straightforward systematic extension to even higher order terms, as commutators involving six or more fermionic operators can be broken down into sums of elementary two-point contractions and dealt with as above.

Similarly, the third part of Eq. 3 can be written as:

$$iii)\left[\mathcal{H}^{(2)}, V^{(4)}\right] = \quad V^{(4)}_{ljkq}\mathcal{H}^{(2)}_{il} + V^{(4)}_{ijlq}\mathcal{H}^{(2)}_{kl} - V^{(4)}_{ilkq}\mathcal{H}^{(2)}_{lj} - V^{(4)}_{ijkl}\mathcal{H}^{(2)}_{lq}$$

$$\qquad\qquad\qquad\qquad\qquad\qquad i \qquad\quad j \qquad\qquad k \qquad\quad q$$

In the end this leads to an anti-Hermitian generator of the form:

$$\eta = \quad \eta^{(2)} \quad + \quad \eta^{(4)}$$

$$\qquad\qquad i\ j \qquad\quad k\ q\ l\ m$$

We can then compute the flow of the Hamiltonian order-by-order using the standard flow equation for $\mathcal{H}$:

$$\frac{\mathrm{d}\mathcal{H}}{\mathrm{d}l} = [\eta, \mathcal{H}] = \left[\eta^{(2)}, \mathcal{H}^{(2)}\right] + \left[\eta^{(4)}, \mathcal{H}^{(2)}\right] + \left[\eta^{(2)}, \mathcal{H}^{(4)}\right] + \dots, \qquad (12)$$

which we can evaluate in exactly the same way as we computed $\eta$ above, using tensor contractions. The ellipsis $(\dots)$ represent terms containing more than four fermionic operators which will be generated by the flow. The lowest order neglected term will be obtained by contracting $\mathcal{H}^{(4)}$ with $\eta^{(4)}$ to generate a new term $\mathcal{H}^{(6)}$, which should then be added back into the ansatz for the running Hamiltonian, which will then in turn lead to another new term $\mathcal{H}^{(8)}$, and so on. The source term for the lowest order neglected term, $\mathcal{H}^{(6)}$ comes from $[\eta^{(4)}, \mathcal{H}^{(4)}] = [[\mathcal{H}^{(4)}, V^{(2)}] + [\mathcal{H}^{(2)}, V^{(4)}], \mathcal{H}^{(4)}]$, therefore it is at most quadratic in the microscopic interaction strength. For weak interactions, each of the newly generated terms will be smaller than the previous order. In the following, we shall assume that we work with sufficiently weak interactions that all newly-generated terms above fourth order are negligible; we will later on examine the validity of this assumption.

At this point, we proceed to numerically solve Eq. 12 directly. In contrast to previous work using flow equations to study disordered many-body systems [42], here we do not analytically write down explicit equations of motion for the coupling constants, but instead generate the flow of the couplings numerically using the above tensor contraction procedure. This has the significant advantage that it avoids a great deal of algebraic complexity and is far more systematic and transparent, allowing us to more easily incorporate higher-order terms and even extend this formalism further than in the present work, something which was extremely challenging in our previous formulation of a flow equation technique [42].

### 3.3 Numerical Procedure

Here we outline the specific steps required to diagonalize a Hamiltonian using the above method. In this work we will always work in real space as we are primarily interested in (quasi)-disordered systems, but it should be noted that this procedure may be used for momentum-space Hamiltonians as well, which may provide advantages for systems where the momentum is a well-defined quantum number.

1. Construct the Hamiltonian $\mathcal{H}$ order-by-order as an $L \times L$ matrix plus an $L \times L \times L \times L$ tensor, i.e. storing $L^2 + L^4$ floating point numbers in memory. This memory storage could be lowered significantly by using symmetries, e.g. $J_{ij} = J_{ji}$ but most linear algebra packages require the full tensor to be constructed before they can be efficiently multiplied, which requires a lot of CPU time to rebuild at every flow time step. If sufficient memory is available, it is faster to store the full matrices/tensors rather than a reduced representation of them. This is an area where there is scope for technical improvement which may facilitate the study of larger system sizes.

2. Compute the generator $\eta = [\mathcal{H}_0, V]$ order-by-order using efficient linear algebra packages to do the tensor contractions specified above, also storing $\eta$ as $L^2 + L^4$ floating point numbers in memory.

3. Compute the right hand side (RHS) of the flow equation $[\eta, \mathcal{H}]$ by performing the tensor contractions specified above.

4. Use a suitable numerical integration routine to integrate from $\mathcal{H}(l) \to \mathcal{H}(l + dl)$ with the RHS as constructed above. Here we make use of the JAX package [82], employing the ode routine with a Runge-Kutta 4th order algorithm and an adaptive step size. When storing the transform, we use a logarithmic grid in flow time, since most of the fastest changes occur at the very start of the flow and the flow-time dynamics slow significantly as $l$ becomes large. This logarithmic grid allows us to more efficiently store the full transform in memory, if necessary, as we focus on storing the greatest number of points in the region where the couplings vary quickly.

5. Repeat until all off-diagonal elements decay ($l \to \infty$ limit). In practice, we numerically integrate to some large finite value of $l$. At each step we consider any couplings smaller than some threshold $\varepsilon$ to be zero, and typically we use $\varepsilon = 10^{-3}$. When all couplings - or at least the majority of them - decay below this threshold, we stop the transform at some finite (usually large) value of $l$. We have checked and using a smaller cutoff does not lead to noticeably different results, but does however come at greatly increased computational cost - see Appendix B. The use of very small thresholds is only advised if extremely high precision is required, for example computing level statistics where the spacings become exponentially small in the system size.

6. **Note**: if we want to store the full unitary transform (e.g. in case we want to reverse the transform later, which is required for computing local integrals of motion and non-equilibrium dynamics), we have to store $q \times (L^2 + L^4)$ floating point numbers, where $q$ is the total number of flow time steps - for large system sizes, this leads to very large memory requirements, and becomes the primary computational limiting factor of this method. This is not a factor if we only wish to diagonalize the Hamiltonian and do not need to store the reverse transform, as in this case we need only store the Hamiltonian itself at each stage ($L^2 + L^4$ floating point numbers), which is typically much more manageable.

### 3.4 Error Estimation: Invariants of the Flow

As the flow equation method involves a truncation in operator space to a polynomial – rather than exponential – number of Hamiltonian coefficients, it is necessarily an approximate method. For small systems, we can compare the results with numerically exact methods (as shown in Appendix B), however the value of this technique lies in its ability to reach larger system sizes than exact methods can access. It is therefore necessary that we develop some form of self-consistent error estimate which can give us confidence that the results can be relied upon even when comparison with exact numerics is not possible.

We can achieve this goal using so-called invariants of the flow. These invariants are quantities which are unchanged under the action of a unitary transform, and therefore by comparing them at the start and end of the flow (i.e. computing the invariants of the $l = 0$ initial Hamiltonian and the $l \to \infty$ final Hamiltonian) we can get a measure of whether the transform has been applied accurately or whether the neglected terms lead to a violation of the unitarity and hence the introduction of an error.

Here we follow the procedure laid out in previous works [42, 43, 75] and make use of the second invariant of the flow, defined as:

$$I_2(l) = \frac{1}{2^L} \mathrm{Tr}\left[\mathcal{H}(l)^2\right],\tag{13}$$

for a system of size $L$. We are interested in whether this quantity changes throughout the flow, and so we define the following quantity of interest:

$$\delta \mathcal{I}_2 = \left| \frac{\mathcal{I}_2(l=0) - \mathcal{I}_2(l \to \infty)}{\mathcal{I}_2(l=0)} \right|.\tag{14}$$

If $\delta \mathcal{I}_2 = 0$, the the transform is exact and perfectly unitary. If, on the other hand, we find that $\delta \mathcal{I}_2 > 0$, this means that terms not included in our ansatz for the running Hamiltonian have at some point during the flow become significant, and we have 'lost' some information by not including them, leading to a deviation from unitarity and thus an error in the final result. In practice, for any finite value of the microscopic interaction strength $\Delta_0 > 0$, we will find a non-zero value of $\delta \mathcal{I}_2$, and as such the reliability of our results rests upon this value being sufficiently small. We will show results for this in the next section.

## 4 Numerical Results

For all of the following, we consider the Hamiltonian given in Eq. 1 with hopping amplitude $J = 1$, nearest-neighbour interactions $\Delta_0 = 0.1$ and simulate systems of size $L = 36$, averaged over $N_s = 128$ disorder realizations. Throughout this work we will use open boundary conditions. We consider three different quasiperiodic potentials $h_i = W \cos(2\pi i/\phi + \theta)$

each generated by a different choice of incommensurate ratio $\phi$ corresponding to the golden ($\phi = (1 + \sqrt{5})/2$), silver ($\phi = 1 + \sqrt{2}$) and bronze ratios ($\phi = (3 + \sqrt{13})/2$). These metallic means can all be generated from the recurrence relation $a_n = m a_{n-1} + a_{n-2}$, where $m$ is an integer, by computing the ratio $a_n/a_{n-1}$ in the limit of $n \to \infty$. For $m = 1$ this gives the golden ratio, and the series $a_n$ is given by the Fibonacci numbers $F_n$, while for $m = 2$ one obtains the silver ratio and for $m = 3$ the bronze ratio. This sequence of metallic means can be continued indefinitely to larger integer values of $m$.

Under the action of the Wegner generator, the initially short-ranged Hamiltonian will acquire long-range hopping and interaction terms over the course of the flow (the former of which will decay to zero in the $l \to \infty$ limit), and as such we make the following ansatz for the scale-dependent running Hamiltonian:

$$\mathcal{H}(l) = \sum_i h_i(l) : n_i : + \sum_{ij} J_{ij}(l) : c_i^\dagger c_j : + \sum_{ij} \Delta_{ij}(l) : n_i n_j : + \sum_{ijkq} \Gamma_{ijkq}(l) : c_i^\dagger c_j c_k^\dagger c_q :, \quad (15)$$

where, as before, we assume that for sufficiently weak interactions any newly generated higher-order terms are negligible, and in the final term we have at least $i \neq j$ or $k \neq q$ such that this term is not diagonal in terms of fermionic number operators. We split this running Hamiltonian into diagonal and off-diagonal components, given by:

$$\mathcal{H}(l) = \mathcal{H}_0(l) + V(l), \quad (16)$$

$$\mathcal{H}_0(l) = \sum_i h_i(l) : n_i : + \sum_{ij} \Delta_{ij}(l) : n_i n_j :, \quad (17)$$

$$V(l) = \sum_{ij} J_{ij}(l) : c_i^\dagger c_j : + \sum_{ijkq} \Gamma_{ijkq}(l) : c_i^\dagger c_j c_k^\dagger c_q : . \quad (18)$$

We emphasize that as compared to previous implementations of the flow-equation method [42, 43, 72, 77] we have now introduced an off-diagonal interaction term, encoded in the tensor $\Gamma_{ijkq}(l)$. While this will eventually decay to zero at long flow times its presence throughout the flow will play an important role, as we are going to discuss in the following.

The end result of our diagonalization procedure will be a Hamiltonian of the form:

$$\tilde{\mathcal{H}} = \sum_i \tilde{h}_i \tilde{n}_i + \sum_{i \neq j} \tilde{\Delta}_{ij} \tilde{n}_i \tilde{n}_j + \dots, \quad (19)$$

where the $\dots$ refers to neglected higher order terms, and the tilde means all quantities are given in the $l \to \infty$ basis.

Sample flows of the coefficients $h_i, J_{ij}, \Delta_{ij}$ and $\Gamma_{ijkq}$ are shown in Fig. 1 for a small system to illustrate how these quantities vary throughout the flow. In particular, the variation of all quantities with flow time $l$ is smooth and well-controlled, with the dynamics occurring in two stages: the first stage is at the early flow times where the coefficients all change rapidly, before settling in to a second stage characterized by a slow asymptotic approach to the diagonal Hamiltonian, which by analogy with renormalization group we also refer to as the 'fixed-point Hamiltonian'. In particular, note that during the first stage of the flow, off-diagonal coefficients which are initially zero (e.g. long-range hopping terms) generically become non-zero, but decay back to zero during the course of the flow. The reason for these two distinct stages of the flow is that the initial Hamiltonian contains only nearest-neighbour couplings, and this represents something of an 'unstable fixed point' for the flow equation procedure. The first stage of the process is a rapid flow *away* from this unstable fixed point, which involves the generation of new couplings, before the second stage of the flow takes over to drive the system to the final diagonal fixed point.

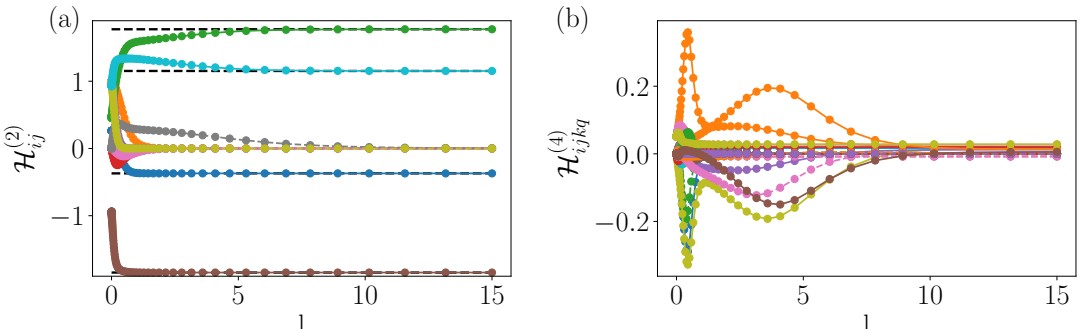

Figure 1: Sample flows of the coefficients in Eq. 15 for a small system of size $L = 4$, with $J = 1$ as the unit of energy, quasi-disorder strength $W/J = 1$ and nearest-neighbour interaction strength $\Delta/J = 0.1$, with incommensurate ratio $\phi = (1 + \sqrt{5})/2$ (the golden ratio) and phase $\theta = 0$. Panel (a) shows the flow of the quadratic terms, $\mathcal{H}_{ij}^{(2)}(l)$, with the on-site terms $h_i(l)$ indicated by the solid lines with circular markers and the off-diagonal terms $J_{ij}(l)$ indicated by dashed lines with square markers. The black dashed lines indicate the single-particle eigenvalues of the system, to which the $\mathcal{H}_{ii}^{(2)}$ terms converge in the $l \to \infty$ limit. Panel (b) shows the flow of the quartic terms $\mathcal{H}_{ijkq}^{(4)}$, again with diagonal terms ($i = j, k = q$) shown as solid lines and off-diagonal terms shown as dashed lines. For reference, by comparison with exact diagonalization the mean relative error in the many-body eigenvalues of this final Hamiltonian is $8.3 \times 10^{-3}$.

One particularly important point to note before moving on is that several of the interaction terms $\Delta_{ij}$ in Fig. 1b) increase dramatically in the early stages of the flow, before eventually decaying back to much smaller values as the flow time $l$ increases. This rapid increase for small values of $l$ is what led to problems with divergent terms encountered in previous works [42,43] when attempting to access delocalized phases at small disorder strengths and/or large interaction strengths. In the present formalism, the inclusion of the off-diagonal quartic terms (denoted $\Gamma_{ijkq}$ in Eq. 15) facilitates this later decay of the density-density interaction coefficients, contrary to what was possible with previous implementations of this technique. In effect, retaining these terms allows them to act like a 'reservoir' for some of the information contained in the initial Hamiltonian, allowing them to become non-zero during the early stages of the flow to 'store' some of this information, before eventually decaying at later stages due to the structure of the generator which guarantees that off-diagonal terms must vanish in the $l \to \infty$ limit.

Now that we have seen an example of the flow equation method in action for a small system where we can visualize the behaviour of the couplings, we will move to much larger systems and will investigate two main quantities of interest:

1. The form of the interactions $\tilde{\Delta}$ in the fixed-point Hamiltonian; in random systems, these decay exponentially with distance, however their real-space form in quasiperiodic systems has not been studied.

2. The real-space support of the LIOM density operators $\tilde{n}_i$; in the diagonal basis, these operators act on a single lattice site, however we can also express them in the initial microscopic basis to extract their real-space support and see how 'local' the LIOMs really are.

Before presenting these results, however, we will briefly introduce our error estimation measures in order to quantify the accuracy of our methods.

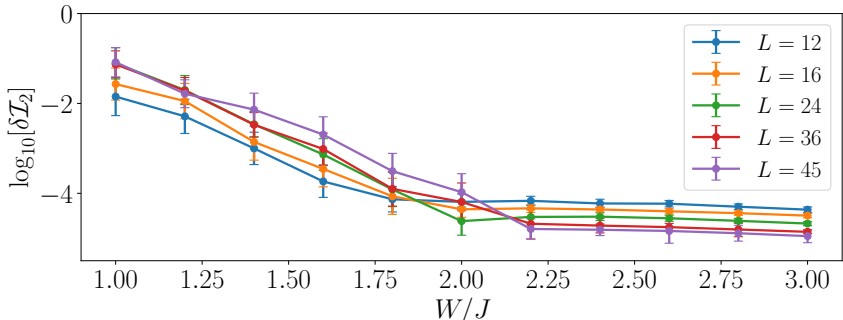

Figure 2: Conservation of the flow invariant, as defined in Eq. 14. Here we take the median value of $\delta\mathcal{I}_2$, over $N_s \in [64, 2048]$ samples, and the error bars show the median absolute deviation. The deviation from unitarity is extremely small in the localized phase where $\delta\mathcal{I}_2$ appears independent of system size. Deep in the delocalized phase ($W/J \ll 2$), there is a slow upwards drift of $\delta I_2$ with increasing system size.

## 4.1 Flow Invariant

First, we will briefly confirm the accuracy of the method by checking the conservation of the flow invariant for the system sizes we will consider in this section. In previous works [42, 43, 75], the flow invariant has been computed from an analytic formula which assumes full diagonalisation is achieved. Here we instead compute the flow invariant numerically. We do so by building the full Hamiltonian in the basis of all two-particle states (using QuSpin), and compute Eq. 14 directly. This works because the trace can be computed for any orthonormal basis, and the basis of all two-particle states is the smallest which takes into account inter-particle interactions, allowing us to compute this measure numerically exactly even for moderate-to-large system sizes.

The results are shown in Fig. 2, averaged over disorder realizations. Here we use the median rather than the mean, as the median is less sensitive to rare individual disorder realizations which require an abnormally long flow time to converge.

We find that the flow invariant is well conserved for all system sizes at large quasi-disorder strengths, with the largest deviations occurring when $W/J$ is small. In this limit, very large flow times are required to fully diagonalize the problem; loosely speaking, the longer the flow runs for, the larger the fourth-order terms in the Hamiltonian become, and correspondingly the larger the neglected terms of sixth-order and higher will also be. In other words, deep in the delocalized phase where $W/J \leq 2$ we may expect the approximate form of the running Hamiltonian to no longer be sufficient, and we see the consequences of this manifest as a deviation from unitary, i.e. a larger value of $\delta\mathcal{I}_2$. At intermediate and large values of $W/J$, however, we see that the flow invariant for all system sizes is conserved to a high degree of accuracy, with deviations on the order of less than 1% for $W/J \geq 2$. Furthermore, the flow invariant in this regime does not display significant variation with system size, again providing a strong indication that our method does not lose accuracy as $L$ increases.

We provide further benchmarks and comparisons with exact diagonalization in Appendix B. Having established the accuracy of the numerical technique, we will now turn to the physics of many-body localization in quasiperiodic systems.

## 4.2 LIOM Interactions

We first investigate the form of the LIOM interactions which appear in the fixed-point Hamiltonian, $\tilde{\Delta}_{ij}$. In particular we will examine how these interactions retain a fingerprint of the

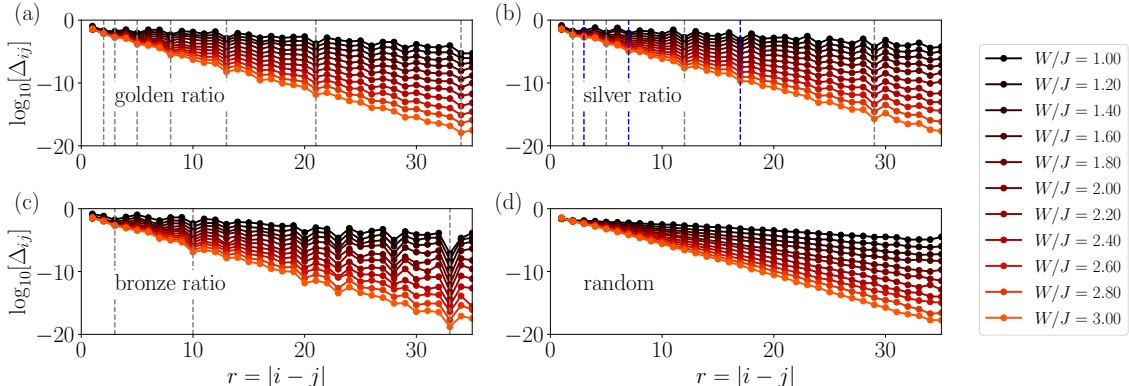

Figure 3: Comparison of the fixed-point Hamiltonian interaction terms for a variety of different potentials: (a) a quasiperiodic potential generated using $\phi = (1 + \sqrt{5})/2$ (the golden ratio), (b) with $\phi = 1 + \sqrt{2}$ (the silver ratio), (c) with $\phi = (3 + \sqrt{13})/2$ (the bronze ratio), and (d) a random potential, as defined in the text. All three quasiperiodic potentials exhibit approximately exponential decay, but with sharp dips at distances $r = |i - j|$ corresponding to numbers from their respective series expansions (grey and blue dashed lines, detailed in the main text). These dips are absent entirely in the case of a random potential, illustrating the how the different structure of the resonances in quasiperiodic systems leads to qualitatively different behaviour from random disorder.

underlying quasiperiodic potential, contrast these results with the LIOM interactions obtained from a random potential, and show how the deterministic distribution of resonances in the quasiperiodic potentials lead to dramatically different results from the case of purely random disorder.

In Fig. 3 we plot the real-space decay of the LIOM interactions, where the [. . . ] notation signifies the median (typical) magnitude across $N_s = 128$ disorder realizations, and we averaged over distances $r = |i - j|$. Panels (a), (b) and (c) refer respectively to the golden ($\phi = (1 + \sqrt{5})/2$), silver ($\phi = 1 + \sqrt{2}$) and bronze ratio ($\phi = (3 + \sqrt{13})/2$), while for comparison we plot in panel (d) the results for the random disorder case. The overall form of the fixed-point interactions is consistent with the exponential decay seen in the random case [42, 43, 83, 84], however there is some additional structure on top of this exponential decay which does not vanish when the number of samples $N_s$ is increased; moreover, the same features are present for all disorder strengths and for all system sizes we have tested ($L \in [12, 64]$).

In particular we see that the LIOM interactions show dips at specific values of the distance $r$. For the golden ratio case in panel (a) we have checked that these dips are located at distances given by the Fibonacci numbers, $r = F_n$, as shown by the dashed lines. It is known from earlier work [31] that for quasiperiodic potentials generated using the golden ratio, single-particle resonances are enhanced at distances $r = F_n$ where $F_n$ is a Fibonacci number. Curiously, the role of these resonances in the many-body problem appears to be the suppression of the LIOM interactions at these particular distances. This observation seems to be in contrast with the suggestion of Ref. [31] where single-particle resonances at distances $r = F_n$ are shown to be enhanced by the quasiperiodic potential, leading to delocalization - naively, one would expect this to manifest as an increase of the LIOM interactions at distances $r = F_n$, rather than the decrease that we observe here. We have checked carefully (see Appendix B) and verified that this effect is not due to convergence errors, and we shall show in Sec. 4.4 that the single-particle component of the LIOMs indeed behaves as indicated by

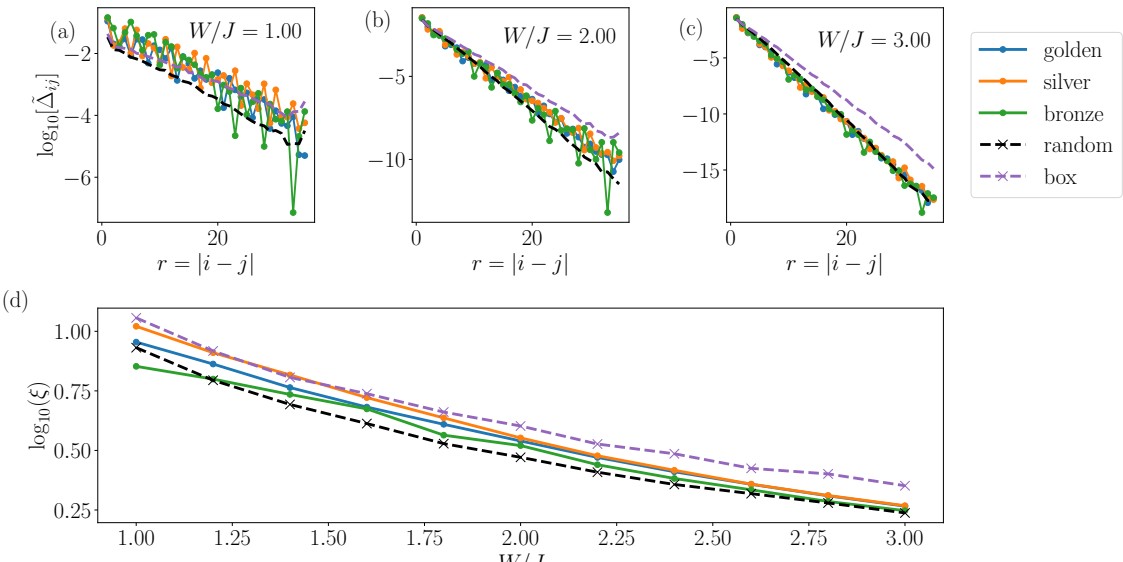

Figure 4: (a-c) Comparison of different potentials at fixed disorder strength. In each case, all three quasiperiodic potentials behave almost indistinguishably (except for the features discussed in Sec. 4.2), the random phase potential (black dashed line) is typically more localized than the quasiperiodic potentials, and the random box potential (purple dashed line) exhibits clear differences for $W/J < 2$ and $W/J > 2$. (d) Localization length computed from each of the different potentials: all follow a qualitatively similar pattern, with the results from the quasiperiodic potential obtained using the bronze ratio displaying a marginally more localized behaviour than the other two quasiperiodic potentials, and the random box system being less localized at large $W/J$.

Ref. [31], but that this suppression at $r = F_n$ again appears (albeit weakly) in the interaction terms of the LIOM operator expansion. Alternatively, it may be that at distances $r = F_n \pm 1$, the LIOM interactions are enhanced by these resonances, leading to the appearance of a dip at $r = F_n$. The structure of local dips appears also for the other metallic means we considered as shown in panels (b) and (c). Analogously to the Fibonacci series, the silver ratio can be approximated by successive ratios of the Pell numbers $P_n = 1, 2, 5, 12, 29 \ldots$, and similarly we see that in Fig. 3b) the fixed-point interactions are suppressed at distances $r = P_n$ (gray lines). Curiously, we also see additional dips at distances equal to half the Pell-Lucas numbers, given by $2, 6, 14, 34 \ldots$ and so on. This arises from the fact that the Pell numbers and Pell-Lucas numbers can be combined to give the closest rational approximations to $\sqrt{2}$: the numerators of the approximations are given by the Pell-Lucas numbers, and the denominators given by the Pell numbers, such that the sequence begins $1/1, 3/2, 7/5, 17/2 \ldots$ with each successive fraction giving a closer approximation to $\sqrt{2}$. Due to the close connection between the silver ratio and $\sqrt{2}$, sites with distances $r$ corresponding to these numbers also exhibit a high degree of spatial correlation, resulting in the additional dips seen in Fig. 3b) (blue lines). As with previous metallic means, also the bronze ratio can be approximated by a series of numbers with the recurrence relation $a_n = ma_{n-1} + a_{n-2}$, here with $m = 3$, which gives rise to the sequence $1, 3, 10, 33, 109 \ldots$. While the rapid increase of this series would suggest that the resonances which arise due to the bronze ratio are typically further apart than for the gold and silver ratios, and that consequently the bronze ratio may lead to more strongly localized behaviour for small systems, in Fig.3b) we in fact see additional dips at distances unrelated to these numbers. Presumably these are linked to other good rational approximations for the

bronze ratio, however we have been unable to verify which approximation they correspond to. Nonetheless, this illustrates the point that the structure of resonances in quasiperiodic potentials is extremely complex.

Finally, in Fig. 3(d) we show the fixed-point couplings $\tilde{\Delta}_{ij}$ for a random potential, with on-site energies $h_i = W \cos(2\pi i/\phi + \theta_i)$ where we again set $\phi = (1 + \sqrt{5})/2$ but crucially in this case we choose the phase to be random on each site, $\theta_i \in [-\pi, \pi]$. We choose this form in order to keep the overall distribution of on-site terms the same between random and quasiperiodic potentials. The dips seen in the quasiperiodic cases are entirely absent here, and instead we observe a precise exponential decay. This confirms that the 'dips' seen previously are entirely due to the deterministic structure of resonances in quasiperiodic potentials, and that for randomly-distributed resonances these features do not appear. we have also checked with a more conventional box distribution of on-site energies (with $h_i \in [-W, W]$, see Fig. 4) and found the same qualitative behaviour. It is also interesting to note that the results obtained for the LIOM interactions bear some similarity with recent strong-disorder renormalization group results on quantum spin chains [85]. In that case it was shown that for quasiperiodic potentials generated by metallic means the couplings would flow to a self-similar binary sequence given by the respective numbers.

As shown in Fig. 4, the fixed-point interactions in the random case are typically more localized than the quasiperiodic systems, which all have an almost identical real-space decay profile. This is due to the spatially correlated nature of the quasiperiodic potentials, which enhances the probability to find resonances between nearby sites, leading to less localized behaviour. In Fig. 4 we also show results for the random potential generated using a box distribution. Here we find that for $W/J > 2$, it seems that Griffiths effects play a significant role, leading to the random box potential exhibiting slightly less localized behaviour. Taken together, the results of studying these different potentials suggest that careful real-space tuning of the resonant sites could be employed to enhance the stability of many-body localization in quasiperiodic potentials without resorting to the commonly studied case of random box disorder, which brings with it rare-region effects which may lead to enhanced delocalization.

### 4.3 Localization Length

As the decay of the fixed-point interactions is approximately exponential in all cases, up to some modulations due to the specific quasiperiodic potential used, we can fit this exponential decay profile to extract a localization length, defined via

$$\left[\tilde{\Delta}_{ij}\right]_{\text{med}} \propto \exp\left(-2\frac{|i-j|}{\xi}\right), \tag{20}$$

where $\xi$ is the localization length, and the 2 in the exponent is from the convention established in studies of Anderson localization, see e.g. Ref. [86]. Although this fitting procedure is somewhat approximate, as it ignores the large deviations in $\tilde{\Delta}_{ij}$ seen when $r = |i-j|$ corresponds to distances typical of single-particle resonances, it is nonetheless instructive.

The results are shown in Fig. 4d) for all three quasiperiodic potentials considered in this work, and the case of random disorder for comparison. We show results for the random potential generated using the cosine-like distribution with a random phase, as in the previous section, and for comparison we also show the results from a random potential generated using a box distribution. We can see that the potentials generated using the golden and silver ratios are very similar, while the potential generated from the bronze ratio is slightly more localized for all values of $W/J$. This confirms the conjecture mentioned in Sec. 4.2 that the more widely separated resonances generated by this potential may lead to more localized behaviour than for the other quasiperiodic potentials studied here, however this effect is very weak. For small $W/J$, the deterministic resonances generated by quasiperiodic potentials dominate, and they

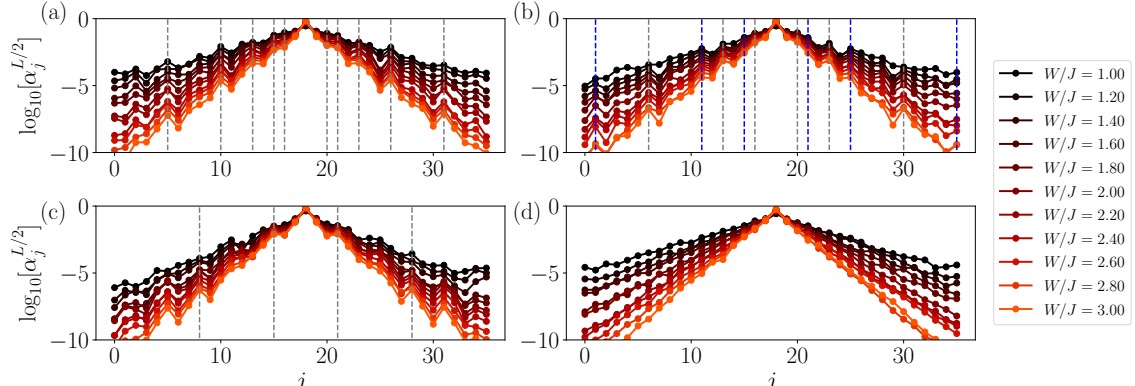

Figure 5: Real space support of a LIOM in the centre of a chain of length $L = 36$, averaged over $N_s = 128$ disorder realizations. Similarly to Fig. 3, the panels show (a) a quasiperiodic potential generated using $\phi = (1 + \sqrt{5})/2$ (the golden ratio), (b) with $\phi = 1 + \sqrt{2}$ (the silver ratio), (c) with $\phi = (3 + \sqrt{13})/2$ (the bronze ratio), and (d) a random phase potential, as defined in the text. The grey and blue dashed lines show the same distances $r$ as indicated in Fig. 3, here measured with respect to the central site on which the LIOM is defined.

lead to delocalization - this can already be seen in the non-interacting Aubry-André model, where the proliferation of these resonances leads to a delocalized phase for $W/J < 2$ which does not exist in the Anderson model. On the other hand, at large $W/J$ the near-resonant sites generated by the quasiperiodic potential are in general too far apart in energy to lead to a proliferation of resonances, and so instead we see that the effects of rare resonant regions in the case of the random box potential win out and lead to less localized behaviour in this regime.

### 4.4 Real-space support of LIOMs

We can also directly compute the real-space support of the local integrals of motion which characterize the fixed point Hamiltonian. Again, the diagonal Hamiltonian is given by:

$$\tilde{\mathcal{H}} = \sum_i \tilde{h}_i : \tilde{n}_i : + \sum_{ij} \tilde{\Delta}_{ij} : \tilde{n}_i \tilde{n}_j : . \tag{21}$$

We can start from the LIOMs in the diagonal basis, given by the local operators $\tilde{n}_i$, and applying the inverse of the unitary transform used to diagonalise the Hamiltonian we can express these operators in the original microscopic basis:

$$\tilde{n}_i = \sum_j \alpha_j^{(i)} : n_j : + \sum_{j \neq k} \beta_{jk}^{(i)} : c_j^\dagger c_k : + \sum_{j \neq k} \gamma_{jk}^{(i)} : n_j n_k : + \sum_{j \neq k \vee l \neq m} \xi_{jklm}^{(i)} : c_j^\dagger c_k c_l^\dagger c_m : + \dots, \tag{22}$$

which we can obtain by numerically solving the equation of motion $dn_i(l)/dl = [\eta(l), n_i(l)]$ backwards from $l \to \infty$ to $l = 0$, with the initial condition $\alpha_j^{(i)}(l \to \infty) = \delta_{ij}$ and all other couplings are zero. This requires that we store the full generator at every stage of the flow, $\eta(l)$, which may consume a large amount of memory for large system sizes.[2] In contrast to previous work, the inclusion of higher-order off-diagonal terms in the running Hamiltonian (Eq. 15) allows us to include higher-order terms in the above ansatz for the density operator.

---

[2]In principle, one could recompute $\eta(l) = [\mathcal{H}_0, V]$ on the fly starting from $\mathcal{H}(l \to \infty)$ and avoid having to store the full transform, however as the new initial conditions for all off-diagonal couplings are all zero, this equation is numerically unstable.

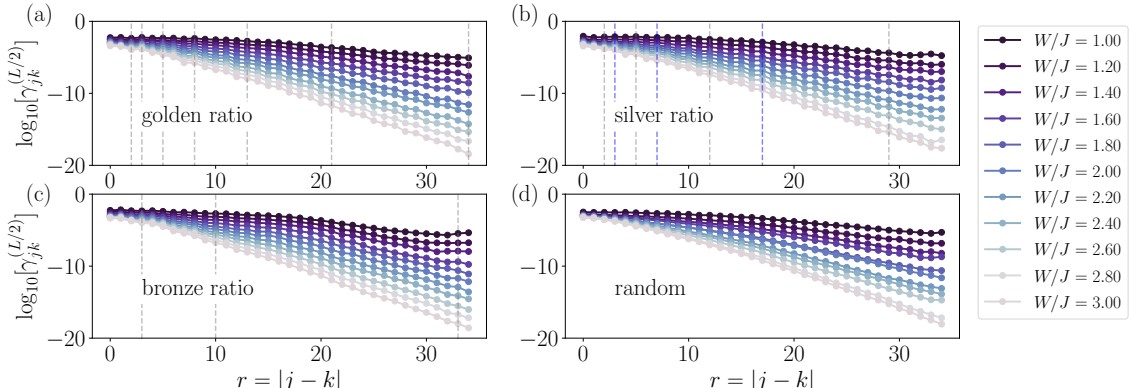

Figure 6: Decay of the 2-body interaction term $\gamma_{jk}^{(i)}$ in Eq. 22 for the central site $i = L/2$ of a system with size $L = 36$, averaged over distance $r = |j - k|$. As in Fig. 3, panel (a) shows results for $\phi = (1 + \sqrt{5})/2$ (the golden ratio), (b) with $\phi = 1 + \sqrt{2}$ (the silver ratio), (c) with $\phi = (3 + \sqrt{13})/2$ (the bronze ratio), and (d) a random potential with on site energies generated using a random phase on each site. Unlike the interaction terms in the effective Hamiltonian $\tilde{\mathcal{H}}$, the 2-body terms here do not have a clear structure on top of the exponential decay, though there are a small number of very weak features at the same distances as highlighted in Fig. 3 (dashed lines).

By looking at the prefactors, we can get some idea of how 'local' these LIOMs really are. We show the results for the lowest-order coefficients $\alpha_j^{(i)}$ in Fig. 5 for a LIOM on the central lattice site ($i = L/2$) and for quasiperiodic potentials given by the golden ratio, silver ratio, bronze ratio and finally for a random potential. Interestingly, the coefficients $\alpha_j^{(i)}$ also exhibit non-monotonic features at distances associated with enhanced resonances, as in Fig. 3, however in this case these distances are associated with local peaks rather than dips. This is in agreement with the results of Ref. [31] who showed that resonances are enhanced at these distances: these enhanced resonances manifest here as peaks in the real-space support of the LIOMs. Interestingly, this same structure can be seen in the real-space behaviour of the eigenstates of the non-interacting Aubry-André model (not shown), suggesting that the quadratic component of the LIOMs inherits this structure from the single-particle eigenstates. As before, this structure is entirely absent in the case of a random potential, shown in Fig. 5d), where the LIOM support decays exponentially with no visible non-monotonic features.

We can also look at the coefficient of the 2-body interaction term in Eq. 22, $\gamma_{jk}^{(i)}$. We show this in Fig. 6, averaged over distance $r = |j - k|$ similarly to the plots of $\tilde{\Delta}_{ij}$ in Fig. 3. The behaviour is very similar to that of the fixed-point interaction coefficients, displaying an approximately exponential decay that becomes steeper at strong (quasi-)disorder strengths. There are hints of some weak additional structure on top of the exponential decay, as seen in previous quantities, however any such features are far weaker than those visible in the fixed point interaction coefficients $\tilde{\Delta}_{ij}$ discussed previously. Again, this structure is absent in the case of the random phase potential, suggesting that its origin is in the deterministic structure of resonances in quasiperiodic systems.

# 5 Evidence for a delocalization phase transition

Having established that we can compute LIOMs in the real-space basis, we can now use the form of these LIOMs to get some insight as to whether there exists a delocalization transition. Although the LIOMs studied in this work are very different to the ones obtained in Ref. [40] (there obtained using exact diagonalization to compute the non-equilibrium dynamics and extract time-averaged local quantities, and here computed by reversing the unitary transform which directly diagonalizes the Hamiltonian - see Ref. [41] for a discussion of this point) the key elements of the analysis can nonetheless be retained.

Following Ref. [40], we define two ratios, $f_2$ and $f_4$, which are the relative weights of the quadratic and quartic terms in the LIOM operator expansion.

$$f_2 = \frac{\sum_j |\alpha_j^{(i)}|^2 + \sum_{jk} |\beta_{jk}^{(i)}|^2}{||n||^2}, \tag{23}$$

$$f_4 = \frac{\sum_{jk} |\Delta_{jk}^{(i)}|^2 + \sum_{jkpq} |\xi_{jkpq}^{(i)}|^2}{||n||^2}, \tag{24}$$

with $||n||^2 = \sum_j |\alpha_j^{(i)}|^2 + \sum_{jk} |\beta_{jk}^{(i)}|^2 + \sum_{jk} |\Delta_{jk}^{(i)}|^2 + \sum_{jkpq} |\xi_{jkpq}^{(i)}|^2$, e.g. $f_2$ is the sum of the squares of the quadratic terms divided by the sum of the squares of all terms, and $f_4$ is the same but for the quartic terms instead. These quantities tell us about the relative weight of the quadratic and quartic terms in the operator expansion, which is related to the role of 2-body collisions and the growth of entanglement. The ability to explicitly compute these $n$-body terms allows us to avoid issues with other observables, such as entanglement or transport dynamics, which may strongly reflect the underlying single-particle critical point at $W_c/J = 2$. Here instead, we are able to explicitly calculate true many-body effects. This is a significant advancement over earlier applications of the flow equation method to disordered systems [42] where only the quadratic terms in Eq. 22 were considered, and the following analysis was not possible.

In Fig. 7 we show both $f_2$ and $f_4$ versus disorder strength $W/J$ for a variety of different system sizes, averaged over $N_s \in [64, 2048]$ disorder realizations depending on system size. Of particular note is that both $f_2$ and $f_4$ display a strong variation with system size for small $W/J$ values that extends up to the largest system sizes we consider here, although this variation slows significantly for the largest systems we simulate, which give virtually identical results. This implies that large system sizes are required in order to obtain accurate behaviour in the delocalized phase, which is in line with previous suggestions that large system sizes and/or simulation times are required to accurately estimate the MBL transition point [87], and illustrates a key advantage of our approach, namely that it allows accurate simulations of large system sizes and provides direct access to the fixed-point Hamiltonian and the local integrals of motion without requiring the dynamical evolution or time-averaging procedure used in other works.

Following the standard procedure for phase transitions, we can make use of a finite size scaling argument to collapse all of the data onto a single curve and extract a crossing point which corresponds to the transition. As in Ref. [40] we find good collapse of data using the scaling form $\Phi(L, W - W_c) = \Phi(L^{1/\nu}|W - W_c|/W_c)$ where $W_c$ is the critical disorder strength and $\nu$ is the critical exponent, which we interpret as evidence that the system undergoes a delocalization transition at a critical value of quasiperiodic potential strength $W_c/J$. We use the Python package `pyfssa` to compute the data collapse and extract these parameters [88, 89].

In Fig. 7, we show the result of the scaling analysis which (approximately) collapses the data onto a single curve and locates a transition point. The scaling of both $f_2$ and $f_4$ give results consistent within their error bars, suggesting that the transition in the interacting system has

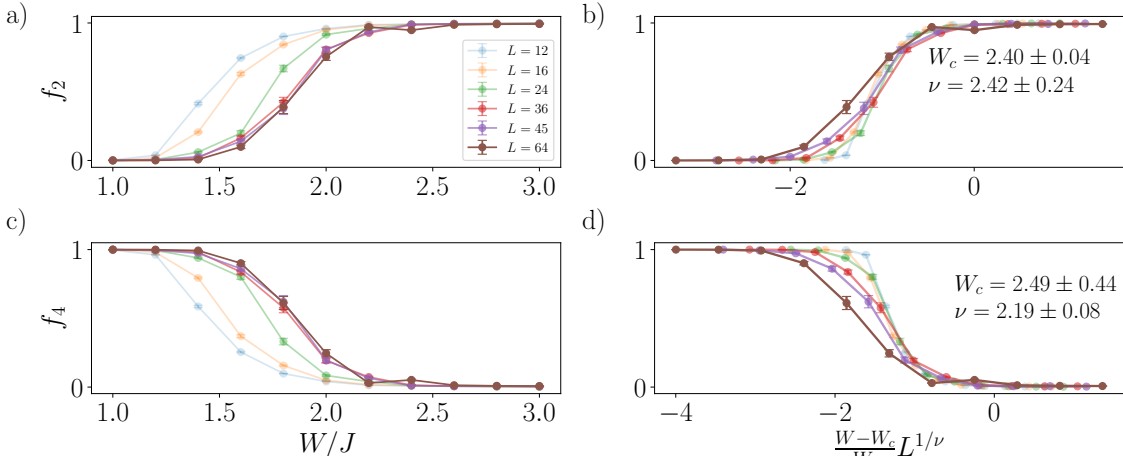

Figure 7: Relative weights of the quadratic ($f_2$) and quartic ($f_4$) terms in the LIOM operator expansion. Panels (a) and (c) show the raw data, including the error bars marking the standard error used as input to the `pyfssa` package. Note that the data for the largest system sizes $L = 36, 45, 64$ are almost on top of one another, suggesting that these system sizes may be large enough to observe the behaviour in the thermodynamic limit. Panels (b) and (d) show the rescaled data, exhibiting good collapse in the region close to $W_c/J \approx 2.4$.

moved to a larger critical disorder strength than in the non-interacting system, as expected from previous work. The uncertainty in the critical value of $W_c/J$ resulting from the data collapse [shown in Fig. 7b) and d)] is smaller than the spacing between data points, indicating that the resolution of our dataset is the dominant source of uncertainty. The values of $W_c/J$ and $\nu$ are both consistent with the results reported in Refs. [39,40], although our lower interaction strength consequently leads to a lower value of $W_c/J$. Interestingly, our results support the claim [39,40] that the critical exponent $\nu$ is well above the Luck bound $\nu \geq 1$ [90] and also fulfills the Harris criterion [91], $\nu \geq 2/d$, thus implying that the quasiperiodic MBL transition is perturbatively stable to the addition of weak randomness.

The central limitation of our results is the quantity of available data for the finite-size scaling, due to the time taken to diagonalize large systems using this method, and in particularly the slow approach to convergence in systems with near-degenerate single-particle eigenvalues (see Appendix B for details on this point), something which is almost guaranteed in large systems with quasiperiodic potential landscapes due to the deterministic structure of resonances which distinguishes quasiperiodic systems. Future developments in increasing the efficiency of the linear algebra operations, and in particular deploying these calculations on clusters of GPUs rather than CPUs, could allow for faster, more accurate data collected on even larger system sizes, which may result in an improved estimate of the critical point which we have tentatively identified in the present work, and may enable a more in-depth study of how this critical point moves with increasing interaction strength.

In the late stages of completion of this work, we became aware of a recent preprint (Ref. [44]) which conducted a careful finite-size scaling analysis of the MBL transition in a quasiperiodic system and concluded that, for system sizes accessible to ED, the best data collapse was achieved with the assumption of a Berezinskii-Kosterlitz-Thouless (BKT) scaling form, as also seen in recent numerical studies on random systems. This result is a surprise in a quasiperiodic system, as the rare ergodic seed regions which lead to avalanches and a BKT-like transition are not currently known to exist in deterministic potentials, and is in contrast with Refs. [27,39,40]. The slow drift of critical disorder strength with increasing system

size interpreted in Ref. [44] as evidence of a BKT-like transition provides further support for the idea that finite-size effects are much weaker in quasiperiodic potentials - in this respect, further studies on large systems will need to be done in order to ascertain whether the most accurate scaling form is the logarithmic drift of system size indicative of a BKT transition, or a more generic system-size-dependent critical disorder which is also shown in Ref. [44] to result in a good collapse of the data. In the present work, we interpret the weak dependence on system size for increasingly large systems as a reflection the slow growth of the number of single-particle resonances at distances $r = F_n$ (where $F_n$ is the $n$th Fibonacci number for $\phi$ equal to the golden ratio) with system size. Here we have assumed the same scaling form as Ref. [40] and found broadly consistent results, however it would be interesting to revisit this assumption in light of the recent results of Ref. [44]. We further suggest a connection to the work of Ref. [54] who found that the localization transition seen in the non-equilibrium dynamics was strongly affected by the choice of potential: based on our results which indicate the importance of the single-particle resonances, we conjecture that the choice of quasiperiodic potential may strongly affect the non-equilibrium dynamics of finite-size systems due to the different distribution of resonances in different quasiperiodic potentials.

## 6    Conclusion

In this manuscript we have studied localization phenomena in a many-body quantum system subject to a quasiperiodic potential, using a new computational implementation of a diagonalization method based around continuous unitary transforms. We have shown that this method represents a significant improvement in accuracy over previous implementations of a truncated flow equation approach, and that the many advances incorporated in this new method allows us to access physics that was out of reach of previous approaches, in particular an estimate of the MBL transition point in a many-body system subject to a quasiperiodic potential via explicit calculation of the local integrals of motion which characterize the system. We have also shown that quasiperiodic potentials result in effective Hamiltonians which contain a strong 'fingerprint' of the structure of the underlying potential, in stark contrast to systems with random disorder which display no such clear feature.

By explicit construction of the local integrals of motion which characterize the MBL phase, we have been able to study their real-space support, gain insight as to how the struction of the LIOMs reflects the underlying potential, and have also used the fraction of quadratic and quartic components of the LIOM operator expansion to tentatively identify a phase transition between localized and ergodic phases. By numerically simulating these quantities for large system sizes, we find critical exponents which are consistent with both large-scale renormalization group studies [39] and recent exact numerics which constructed the LIOMs in a different manner [40]. In particular, we find a critical exponent $\nu > 2/d$, where $d$ is the spatial dimension, implying the perturbative stability of the quasiperiodic MBL transition to weak randomness [91]. Our results for the largest system sizes we have studied are almost identical: as finite-size effects are known to be weaker in quasiperiodic potentials than in true disordered systems, this suggests that we are able to simulate system sizes large enough that our results are an accurate reflection of the thermodynamic limit, and that finite-size effects play only a small role here.

As with all approaches based around diagonalizing a static Hamiltonian, it is important to note that this is not unequivocal evidence for a true phase transition. The MBL transition is a dynamical phenomenon which occurs when highly excited eigenstates spontaneously fail to thermalize: here we have studied the local integrals of motion which characterize the diagonal Hamiltonian, both in terms of their real-space structure and their effective interactions,

however the precise localization properties of any particular initial state will depend on its decomposition in terms of these LIOMs. In other words, a key future goal for this method will be its extension to also transform arbitrary initial states in order to express them explicitly in terms of the LIOMs computed here, in a manner which is efficient and computationally tractable.

In this work we have focused on static properties of the local integrals of motion and the corresponding fixed-point Hamiltonian $\tilde{\mathcal{H}}$, however the method presented can be readily extended to compute the non-equilibrium dynamics following a quench, as in previous works using the flow equation method [42, 43, 77], enabling the study of operator spreading [77] via a decomposition of a local operator similar to that used here (Eq. 22). It is also possible to compute correlation functions using this method [77], which provides an alternative way to define a localization length. The applications of this method go beyond localization in one-dimensional quasiperiodic systems. We have shown in the present manuscript that our results are valid on both sides of the localization transition, in contrast to previous truncated flow equation implementations, and as such this method is also well-suited for the study of more conventional MBL transitions in random systems. As the incorporation of higher-order terms in the ansatz of Eq. 15 is now possible in a systematic manner, it may also be possible to study more strongly interacting systems with high accuracy simply by including more terms in Eq. 15. As all flow equation methods based on Wegner-type generators result in the generation of new long-range couplings throughout the flow, there is essentially no performance cost to including long-range terms in the initial Hamiltonian, meaning that this method is also extremely well-suited to the study of localization in systems with long-range couplings, e.g. building on the results of Ref. [43]. The lack of explicit dependence on the geometry of the system similarly means that an $L = 64$ chain, for example, can be diagonalized with essentially the same complexity as an $L = 8 \times 8$ two-dimensional system, facilitating the construction of local integrals of motion in two-dimensional systems. The method is not restricted to fermions either, and can be generalized to bosonic systems simply by taking into account bosonic rather than fermionic commutation relations: this may allow an advantage over tensor network based methods where the size of the local bosonic Hilbert space features explicitly in the construction of matrix product state wavefunctions, which can quickly become prohibitive for weakly-interacting systems where large local occupations are possible. Here, by contrast, the dimension of the local Hilbert space does not enter explicitly in the diagonalization nor the construction of the LIOMs.

The biggest computational advantage of this method, however, is the significant speed increase that can be obtained from using graphical processing units (GPUs) rather than the more conventional CPUs used in the majority of current numerical works in many-body physics. As we demonstrate in Appendix C, for large systems even a modest GPU can result in a significant speed increase. In the present manuscript, we have taken only the first steps in this direction: we expect that the continued development of this method on cutting-edge GPU hardware will enable its extension into parameter regimes which are currently inaccessible to all other theoretical methods. For completeness, and to motivate others to adopt flow equation methods, we also present a small sample code in Appendix D illustrating the application of this technique to a non-interacting system.

## Acknowledgements

This project has received funding from the European Union's Horizon 2020 research and innovation programme under the Marie Skłodowska-Curie grant agreement No. 101031489 (Ergodicity Breaking in Quantum Matter), as well as support from the ANR grant "NonEQuMat"

(ANR-19-CE47-0001). SJT also acknowledges support from an NVIDIA Academic Hardware Grant. Initial computations were performed on the Collège de France IPH-JE computing cluster. We gratefully acknowledge useful discussions with B. Ware and feedback on the manuscript from L. Sanchez-Palencia, as well as valuable comments from C. Bertoni. Code and data are available at [56, 92].

# A  Consistent ordering of operators

In the main text, we have assumed a consistent order of fermionic operators in the Hamiltonian, however we did not go into detail about how this is achieved. In general swapping the order of two fermionic operators can result in the generation of new terms, e.g.:

$$c_i^\dagger c_j c_k^\dagger c_q = c_i^\dagger (-c_k^\dagger c_j + \delta_{jk}) c_q = -c_i^\dagger c_k^\dagger c_j c_q + \delta_{jk} c_i^\dagger c_q \,. \tag{A.1}$$

Therefore if this term arose when evaluating $\mathrm{d}\mathcal{H}/\mathrm{d}l$, one could arrange the operators in the form $c_i^\dagger c_j c_k^\dagger c_q$ and conclude that this term renormalizes the quartic terms only, or one could rearrange the operators into the form $-c_i^\dagger c_k^\dagger c_j c_q + \delta_{jk} c_i^\dagger c_q$ and conclude that this term renormalized both quartic and quadratic parts of the Hamiltonian. While not necessarily inconsistent, this arbitrariness complicates the computation of high-order commutators. The key ingredient in avoiding this ambiguity is known as normal-ordering, which enforces a consistent ordering of the operators in each term. A detailed description of normal ordering in the context of unitary flow methods is given in Refs. [58, 93], and we briefly summarize the main points here. While this prescription is often thought of as simply subtracting the expectation value from an operator, the fundamental idea is rather more sophisticated. For fermions, it consists of expressing a fermionic anticommutation relation as:

$$\{c_i^\dagger, c_j\} = G_{ij} + \tilde{G}_{ji} \,, \tag{A.2}$$

where we define the contractions used in normal-ordering as:

$$G_{ij} = \langle c_i^\dagger c_j \rangle \,, \tag{A.3}$$

$$\tilde{G}_{ji} = \langle c_j c_i^\dagger \rangle \,, \tag{A.4}$$

and the expectation values are computed in some reference state, which is typically the ground state of the corresponding non-interacting model, but can also be a more complex mixed state described by some density matrix. To calculate the commutation relations of normal-ordered strings of operators, we use the following theorem [58]:

$$: O_1(A) :: O_2(A') := \, : \exp\left( \sum_{kl} G_{kl} \frac{\partial^2}{\partial A'_l \partial A_k} \right) O_1(A) O_2(A') : \,, \tag{A.5}$$

where $A$ and $A'$ are the set of labelled operators in the expression $O_1$ and $O_2$ which in our case are just strings of fermionic operators. We can evaluate this by expanding the exponential - for simple expressions, the expansion can be exact, as all derivatives are zero above some order greater than the number of operators in $O_1$ or $O_2$.

   The precise choice of reference state in which the contractions are computed depends on the calculation in question: in Refs. [42, 43], we used a uniformly half-filled product state with $\langle n_i \rangle = 0.5 \forall i$ to approximate an arbitrary state of high energy density, although in the framework of that calculation the precise state we chose made little difference: incorporation of normal ordering corrections was only necessary in order to prevent divergent interacting

terms at small disorder strengths, as in Refs. [42,43] the main role of normal-ordering corrections was to terminate the flow early if interaction terms began to diverge. Here, however, due to the more sophisticated form of running Hamiltonian which we employ, these divergences are no longer an issue, and we are free to choose a simpler form of normal ordering, which we have verified the accuracy of via a series of numerical checks B. Here we use vacuum normal ordering, commonly thought of as 'moving all dagger operators to the left' in any composite string of fermionic operators. Vacuum normal ordering implies that $G_{ij} = 0$ and $\tilde{G}_{ji} = \delta_{ij}$, such that we still have the general identity $G_{ij} + \tilde{G}_{ji} = \delta_{ij}$ that satisfies fermionic anticommutation relations.

### A.1 Example: Quadratic terms

$$: c_\alpha^\dagger c_\beta :: c_\gamma^\dagger c_\delta := \left( 1 + G_{\alpha\delta} \frac{\partial^2}{\partial c_\alpha^\dagger \partial c_\delta} + \tilde{G}_{\beta\gamma} \frac{\partial^2}{\partial c_\beta \partial c_\gamma^\dagger} + G_{\alpha\delta}\tilde{G}_{\beta\gamma} \frac{\partial^4}{\partial c_\alpha^\dagger \partial c_\delta \partial c_\beta \partial c_\gamma^\dagger} \right) c_\alpha^\dagger c_\beta c_\gamma^\dagger c_\delta : \quad \text{(A.6)}$$

$$=: c_\alpha^\dagger c_\beta c_\gamma^\dagger c_\delta : + G_{\alpha\delta} : c_\beta c_\gamma^\dagger : + \tilde{G}_{\beta\gamma} : c_\alpha^\dagger c_\delta : + G_{\alpha\delta}\tilde{G}_{\beta\gamma} . \quad \text{(A.7)}$$

So the commutation relation is given by:

$$[: c_\alpha^\dagger c_\beta :, : c_\gamma^\dagger c_\delta :] = -(G_{\alpha\delta} + \tilde{G}_{\alpha\delta}) : c_\beta c_\gamma^\dagger : + (\tilde{G}_{\beta\gamma} + G_{\gamma\beta}) : c_\alpha^\dagger c_\delta : + (G_{\alpha\delta}\tilde{G}_{\beta\gamma} - G_{\gamma\beta}\tilde{G}_{\delta\alpha}) \quad \text{(A.8)}$$

$$= \delta_{\beta\gamma} : c_\alpha^\dagger c_\delta : - \delta_{\alpha\delta} : c_\gamma^\dagger c_\beta : + (G_{\alpha\delta}\tilde{G}_{\beta\gamma} - G_{\gamma\beta}\tilde{G}_{\delta\alpha}), \quad \text{(A.9)}$$

which, after applying the vacuum normal-ordering identity $G_{ij} = 0$, reduces back to the same result as one would obtain without normal-ordering.

### A.2 Example: Higher-order terms

The higher-order terms proceed in essentially the same way, giving:

$$
\begin{aligned}
[: c_\alpha^\dagger c_\beta c_\gamma^\dagger c_\delta :, : c_\mu^\dagger c_\nu :] = \quad & -(G_{\alpha\nu} + \tilde{G}_{\nu\alpha}) : c_\mu^\dagger c_\beta c_\gamma^\dagger c_\delta : -(G_{\gamma\nu} + \tilde{G}_{\nu\gamma}) : c_\alpha^\dagger c_\beta c_\mu^\dagger c_\delta : \\
& +(\tilde{G}_{\beta\mu} + G_{\mu\beta}) : c_\alpha^\dagger c_\nu c_\gamma^\dagger c_\delta : +(\tilde{G}_{\delta\mu} + G_{\mu\delta}) : c_\alpha^\dagger c_\beta c_\gamma^\dagger c_\nu : \\
& +(G_{\alpha\nu}\tilde{G}_{\beta\mu} - G_{\mu\beta}\tilde{G}_{\nu\alpha}) : c_\gamma^\dagger c_\delta : +(G_{\alpha\nu}\tilde{G}_{\delta\mu} - G_{\mu\delta}\tilde{G}_{\nu\alpha}) : c_\gamma^\dagger c_\beta : \\
& +(G_{\gamma\nu}\tilde{G}_{\beta\mu} - G_{\mu\beta}\tilde{G}_{\nu\gamma}) : c_\alpha^\dagger c_\delta : +(G_{\gamma\nu}\tilde{G}_{\delta\mu} - G_{\mu\delta}\tilde{G}_{\nu\gamma}) : c_\alpha^\dagger c_\beta : . \quad \text{(A.10)}
\end{aligned}
$$

After applying vacuum normal-ordering identities for the contractions, this results in:

$$
\begin{aligned}
[: c_\alpha^\dagger c_\beta c_\gamma^\dagger c_\delta :, : c_\mu^\dagger c_\nu :] = & -\delta_{\alpha\nu} : c_\mu^\dagger c_\beta c_\gamma^\dagger c_\delta : -\delta_{\gamma\nu} : c_\alpha^\dagger c_\beta c_\mu^\dagger c_\delta : \quad \text{(A.11)} \\
& +\delta_{\beta\mu} : c_\alpha^\dagger c_\nu c_\gamma^\dagger c_\delta : +\delta_{\delta\mu} : c_\alpha^\dagger c_\beta c_\gamma^\dagger c_\nu :,
\end{aligned}
$$

which is exactly the same result as obtained from the (much simpler) graphical notation introduced in Sec. 3.2. Even higher order terms may be computed in similar ways, though we do not go into further detail here.

## B Checks

### B.1 Comparison with ED

As a first check of our method, for small systems we benchmarked the many-body eigenvalues obtained using the flow equation (FE) approach against numerically exact diagonalization

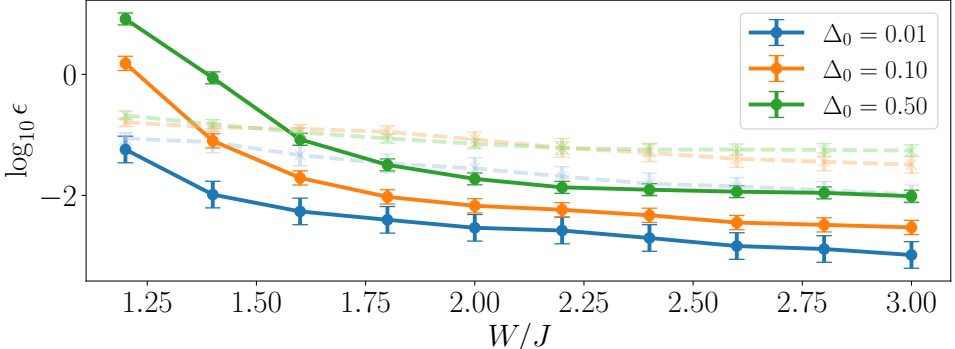

Figure 8: Median relative error $\epsilon$ in the many-body eigenenergies obtained from the flow equation method for $L = 12$ and three different microscopic interaction strengths $\Delta_0$, using $\phi = (1 + \sqrt{5})/2$ (the golden ratio) and averaged over $N_s = 2048$ random values of the phase $\theta$, computed with respect to the results obtained from exact diagonalization. The error bars represent the median average deviation over different random values of the phase $\theta$. The dashed lines represent the result of the older implementation used in Refs [42, 43].

(ED) using the QuSpin library [94,95]. Here we present some of these results for small systems in order to demonstrate the accuracy of our method. In the diagonal $l \to \infty$ basis, the many-body eigenstates are simply product states of LIOMs, e.g. $|\psi\rangle = |0001, 0010, 0011 \ldots\rangle$, and can be straightforwardly constructed by applying the Hamiltonian $\tilde{\mathcal{H}}$ to all $2^L$ possible product states. Note that this is a very demanding check of our method, as it involves *all* many-body eigenstates, not simply the half-filled states or those in the middle of the excitation spectrum. Once the eigenvalues $E_n$ are obtained, the relative error in the $n$-th eigenvalue is defined as:

$$\epsilon_n = \left| \frac{E_n^{ED} - E_n^{FE}}{E_n^{ED}} \right|, \tag{B.1}$$

where the superscripts $ED$ and $FE$ refer to the eigenvalues obtained with exact diagonalization and flow equation methods respectively. The results are shown in Fig. 8 for system size $L = 12$ and three different interaction strengths, each in a quasiperiodic potential generated using the golden ratio and averaged over $N_s = 2048$ values of random phase $\theta$. In the localised phase, the median relative error is extremely small, but diverges at low values of $W/J$ when the system becomes delocalised. We note that the relative error computed for the same system using the older method of Refs. [42,43] (dashed lines in Fig. 8) is larger in the localised phase, but smaller in the delocalised phase: this is a combination of the normal-ordering procedure used in Ref. [43] to prevent divergences, and the fact that fewer terms were kept in the older implementation, resulting in fewer divergent terms and correspondingly a smaller error.

## B.2 Convergence checks

To evaluate the performance of diagonalization methods based on continuous unitary transforms, one key metric is the how quickly the transform converges. For the Wegner generator used in the present manuscript, the convergence is determined by how widely-separated the single-particle eigenvalues are, with more widely separated eigenvalues leading to faster convergence, which implies that in general the convergence will be faster for larger quasi-disorder amplitudes - see Refs. [42,43,58,77] for further discussion of this point. We first examine the convergence properties of the non-interacting Aubry-André model before looking at the convergence of the interacting system given in Eq. 1.

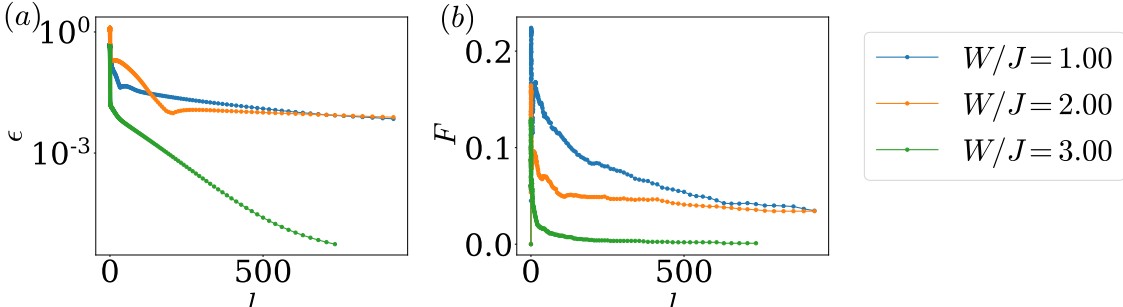

Figure 9: (a) Mean relative error in the diagonal matrix elements of the non-interacting Aubry-André model throughout the flow, computed with respect to the exact single-particle eigenvalues for a system of size $L = 64$ for $\phi = (1 + \sqrt{5})/2$ (the golden ratio), a single value of the phase ($\theta = 0$). Note that although the data for small values of $W/J$ may appear flat on this scale, they are in fact decaying exponentially with a shallow gradient. (b) Fraction of off-diagonal couplings at each stage in the flow which are above the threshold cutoff of $10^{-3}$. There is an initial increase as new couplings are generated in the first few flow time steps, followed by a sharp decrease at small flow times, and then a slow decay towards zero.

To give some idea of the flow times required in order to obtain the eigenvalues to a given accuracy, in Fig. 9a) we show the relative error $\epsilon$ of the on-site terms of a non-interacting system, computed with respect to the exact single-particle eigenvalues obtained by directly diagonalizing the matrix, while in Fig. 9b) we show the fraction $F$ of off-diagonal matrix elements which are above the cutoff of $10^{-3}$.

Here we use a system size $L = 64$ and three different quasi-disorder strengths, with each quasi-disorder potential representing a single random realization obtained using the golden ratio. We integrated to a maximum flow time $l_{max} = 10^3$, but stop the flow early if all off-diagonal elements have decayed below the cutoff (as seen for $W/J = 3$). For all values of $W/J$ the error exhibits a sharp decrease at the beginning of the flow, with larger values of $W/J$ typically decaying more quickly, before exhibiting a slow exponential decay towards zero in the later states of the flow. For quasi-disorder realizations with near-degenerate eigenvalues, this flow can be extremely slow and require very large flow times in order to obtain convergence to a high accuracy. We note that the late-time convergence is slowest around the critical point of the non-interacting system $W/J = 2$ where the spectrum is fractal in nature [47] and therefore the eigenvalues are very close together. As we are primarily interested in the range $W/J > 2$, in the main text we used a maximum flow time of $l_{max} = 25$, which we found to be a good compromise between speed and accuracy, with a typical error in the single-particle eigenvalues of around $\epsilon \lesssim 10^{-2}$. Beyond this point we see diminishing returns by going to larger flow times, and in any case relative error in the presence of interactions is primarily limited by the truncated form of Eq. 15 and the strength of the microscopic interactions rather than the maximum flow time, so we do not find larger flow times to lead to a meaningful increase in accuracy.

We now turn to the convergence of the interaction terms. By comparison with the findings of Ref. [31] who noted that single-particle resonances are enhanced in quasiperiodic systems generated using the golden mean at distances corresponding to the Fibonacci numbers $F_n$, one might ask whether the 'dips' seen in the fixed-point couplings $\tilde{\Delta}_{ij}$ discussed in Sec. 4.2 could be due to convergence errors, i.e. the associated hopping terms $J_{ij}$ may not have decayed sufficiently under the action of the transform, leading to an incorrect result for $\tilde{\Delta}_{ij}$ at these distances. In Fig. 10 we show that this is not the case, and that on the contrary we do not

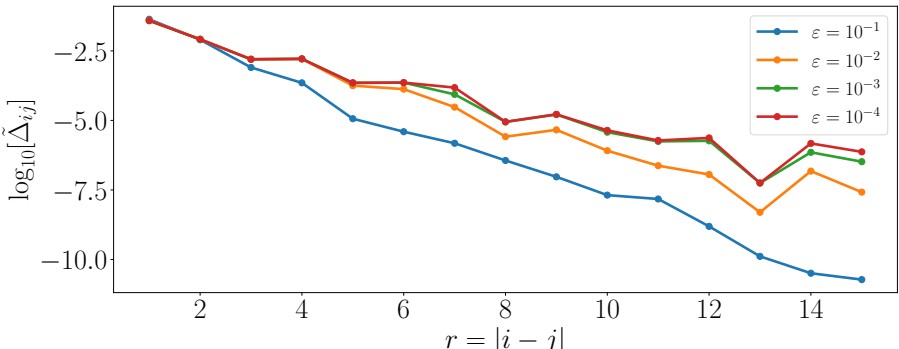

Figure 10: Convergence of the interaction terms in the running Hamiltonian, $\tilde{\Delta}_{ij}$, shown for a system of size $L = 16$, $\phi = (1 + \sqrt{5})/2$ (the golden ratio), one single value of the phase ($\theta = 0$), and several different values of the cutoff $\varepsilon$. In each case, the flow is run until *all* off-diagonal elements have decayed to below the cutoff value. We see that for large cutoffs the $\tilde{\Delta}_{ij}$ terms decay approximately exponentially with distance, but when smaller cutoffs are used, the characteristic 'dips' at distances $r = F_n$ appear, where the $F_n$ are the Fibonacci numbers. This confirm that these features are not convergence errors.

see these dips until all couplings have sufficiently decayed, here for approximately $\varepsilon \leq 10^{-2}$ of their initial value. For still smaller cutoffs there are some quantitative changes, however these features persist. In the results presented in the main manuscript, we use a flow time such that the majority of couplings have decayed below $\varepsilon = 10^{-3}$: we do not insist that *all* couplings decay below this threshold, as in this case our simulations become limited by unusual realizations of the quasiperiodic potential where near-degeneracies can cause a small number of couplings to flow extremely slowly. Loosely speaking, requiring all off-diagonal elements to decay by another order of magnitude requires a flow time (and therefore computation time) which is itself an order of magnitude larger, and this quickly becomes prohibitive once we go beyond $\varepsilon = 10^{-3}$. While it may be the case that additional effects not taken into account in the present calculation, e.g. higher-order terms or an alternative form of normal-ordering, could modify our results, the features seen in the fixed-point couplings $\tilde{\Delta}$ seem robust and it is hard to anticipate any modifications that would destroy them.

### B.3 Comparison with previous implementations of flow equations

An earlier implementation of continuous unitary transforms for the study of MBL phenomena was developed in Refs. [42, 43] using a more approximate form of the running Hamiltonian which allowed us to analytically obtain flow equations for the running couplings, but which by construction did not allow for the generation of off-diagonal quartic terms during the flow - we refer the reader to Refs. [42, 43] for further details regarding this earlier implementation. The numerous and varied improvements presented in this manuscript lead to a significant improvement over the older method, which we shall demonstrate here.

In Fig. 11 we show the results for the fixed-point couplings $\tilde{\Delta}_{ij}$ obtained using both the old and new flow equation implementations, both for the same system size of $L = 36$ and averaged over $N_s = 128$ realizations of the quasiperiodic potential generated using the golden ratio, as in Fig. 3a). There is a clear difference between the results, particularly at short distances. Due to the inclusion of higher-order off-diagonal terms, the short-range behaviour is captured much more accurately by the new implementation: in the previous implementation, the short-range couplings can exhibit divergences at small values of the quasiperiodic potential amplitude (i.e.

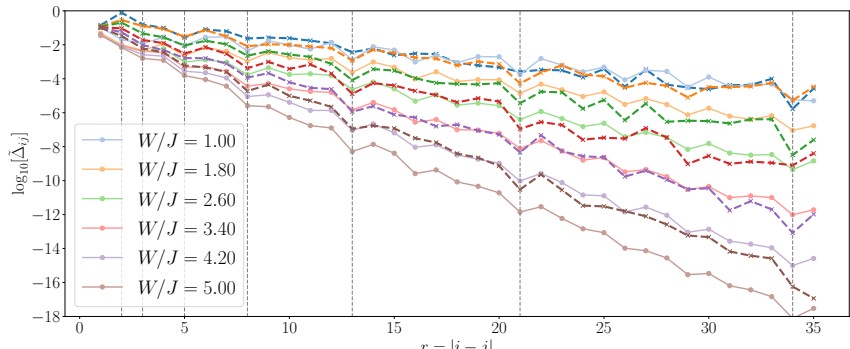

Figure 11: Comparison of the truncated flow equation method from Refs. [42, 77] compared with the method presented in this manuscript. Here we reproduce the results from Fig. 3a) [solid lines], and compare with the same quantity computed using the older flow equation implementation [dashed lines]. We can clearly see that while the broad features are the same, the new method leads to significantly modified short-range behaviour, and consequently far more localized fixed-point interactions $\tilde{\Delta}_{ij}$.

in the delocalized phase). The new implementation does not exhibit the same problem, and this change in the short-range behaviour leads to a noticeably faster decay of the couplings, corresponding to a 'more local' unitary transform. Nonetheless, at longer distances many of the main features present in the results obtained using the new method are also present in the older implementation, notably the 'dips' at distances $r = F_n$ where the $F_n$ are the Fibonacci numbers.

## C  CPU/GPU Speed Comparison

All comparisons in this Appendix were performed on a standard laptop, a Dell G5 5590 with a 6-core (12-thread) Intel Core i7 9750-H CPU with 16Gb RAM and a 6Gb NVIDIA GTX1660 Ti GPU. Even with modest hardware, the speed increase from making use of the high degree of parallelization possible with a GPU can be significant. These comparisons made use of the `PyTorch` library [96] and the `torchdiffeq` package [97,98] to facilitate the straightforward use of sophisticated integration routines on a GPU: to facilitate a like-for-like comparison, precisely the same code was run on the CPU by setting `device='cpu'` in the `PyTorch` settings. This is in contrast to the code used in the main part of this manuscript which used custom-coded tensor contraction routines using `Numba`'s just-in-time (JIT) compilation feature to achieve the fastest possible performance [99]. Here we use a flow time $l_{max}$ large enough that all couplings have decayed below the threshold of $\varepsilon/J = 10^{-3}$, and all code uses single-precision floats. We used a quasiperiodic potential generated using the golden ratio, and fixed the phase to be $\theta = 0$ for all simulations. The diagonalization for each system size was run several times (except for $L = 24$, for which we conducted only one run due to the long time required on the CPU) and then we took the average value of the total time taken: error bars showing the standard deviation over different runs are smaller than the plot markers. These benchmarks show the time taken to fully diagonalize the Hamiltonian and to transform a single local operator from the initial basis to the final diagonal basis. The time shown is the full execution time of the script, measured using the `dateTime` module.

As shown in Fig. 12, the relative speed of the GPU over the CPU increases sharply with system size, with simulations for $L = 24$ which took approximately 2 hours on the CPU taking

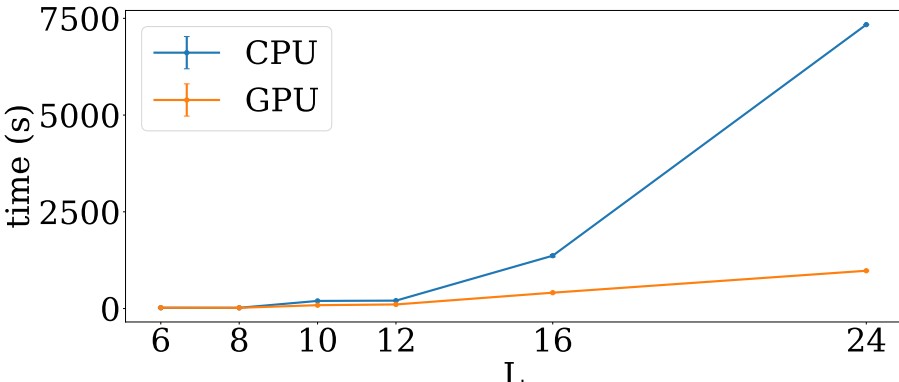

Figure 12: Comparison of time taken to diagonalize a system of size $L$ for CPU (blue) and GPU (orange). As the system size increases, the relative speed-up possible with a GPU increases significantly.

around 15 minutes on the GPU. We expect this trend to continue to larger system sizes, with GPUs exhibiting a clear performance advantage for the diagonalization of large many-body systems, and perhaps enabling further studies of MBL phenomena in two-dimensional models where diagonalization with a CPU would be unfeasible. The limiting factor in the utility of GPUs is their relatively small memory, which becomes an issue if one wishes to store the full unitary transform (typically $O(10^3 \times L^4)$ floating point numbers, which can rapidly reach $10^2$ Gb or more in size) in order to be able to reverse the transform from the diagonal basis back into the initial microscopic basis: to achieve this on large system sizes, it may be necessary to develop more sophisticated algorithms which can run on GPU clusters and take advantage of their shared memory, or develop a more efficient way of passing the unitary transform data from the GPU to the CPU during the initial diagonalization, and quickly re-loading it back to the GPU when applying the reverse transform.

## D  Sample Code for Non-interacting Systems

As a simple, non-optimized example to demonstrate the applicability of this method, here we provide a sample Python code for diagonalising a non-interacting system using this technique. Here we prioritize readability over speed, and so the following is adapted from the code used in the main manuscript to be a short, self-contained example with most of the sophisticated features stripped out.

First we import the required modules [100–102] and set the Hamiltonian parameters:

```python
import numpy as np
from scipy.integrate import odeint
import matplotlib.pyplot as plt

dis_type = 'QP'               # Specify potential: 'random' or 'QP'
n = 4                         # System size
d = 2.                        # Quasi-disorder strength
J = 1.                        # Nearest-neighbour hopping strength

np.random.seed()              # Re-seed random number generator

# Generate list of flow time steps to store
dl_list = np.linspace(0,10,100,endpoint=True)
```

For the purposes of this example, we chose an $l_{max}$ that should be sufficiently large to ensure

convergence to the diagonal Hamiltonian, however in the code used to obtain our main results we simply set a large value of $l_{max}$ and stop the flow once all off-diagonal couplings have decayed below some threshold value which we choose to be small.

Next, we initialize the Hamiltonian by defining matrices for the on-site terms $\mathcal{H}_0$ and the off-diagonal terms $V$:

```
1  # Non - interacting matrices
2  H0 = np.zeros ((n,n),dtype=np.float64)
3  if dis_type == 'random':
4      for i in range(n):
5      # Initialise Hamiltonian with random on-site terms
6          H0[i,i] = np.random.uniform(-d,d)
7  elif dis_type == 'QP':
8      phase = np.random.uniform(-np.pi,np.pi) # Random phase
9      phi = (1.+np.sqrt(5.))/2.                # Golden ratio
10     for i in range(n):
11     # Initialise Hamiltonian with quasiperiodic on-site terms
12         H0[i,i] = d*np.cos(2*np.pi*(1./phi)*i + phase)
13
14 # Initialise V0 with nearest-neighbour hopping along leading
       diagonals
15 V0 = np.diag(J*np.ones(n-1,dtype=np.float64),1)
16 V0 += np.diag(J*np.ones(n-1,dtype=np.float64),-1)
```

Now we can define a function to compute the commutator of two matrices, $[A,B] = AB - BA$. Here we use NumPy's `einsum` routine as its inner workings are quite transparent due to the explicit labelling of indices, however `tensordot` is also a good (and often faster) choice, and the best performance can be found by using `Numba` or `Cython` to write compiled code which can be efficiently parallelized:[3]

```
1  # Function to contract square matrices (matrix multiplication)
2  def comm(A,B):
3      return np.einsum('ik,kj->ij',A,B) - np.einsum('ik,kj->ij',B,A)
```

We can also define a function to compute the generator $\eta = [\mathcal{H}_0, V]$ and then the right hand side of the flow equation $d\mathcal{H}/dl = [\eta, \mathcal{H}]$:

```
1  # Function to compute the RHS of the flow equation
2  def nonint_ode(H,l):
3      n= int(np.sqrt(len(H)))
4      H = H.reshape(n,n)
5      H0 = np.diag(np.diag(H))
6      V0 = H - H0
7      eta = comm(H0,V0)
8      sol = comm(eta,H)
9
10     return sol.reshape(n**2)
```

And finally we pass this function to the integrator `odeint` and integrate the differential equation for the Hamiltonian up to our chosen $l_{max}$ value.[4] Note that `odeint` does not accept matrix inputs, so we first flatten the $n \times n$ matrix into a list of length $n^2$.

```
1  sol = odeint(nonint_ode,(H0+V0).reshape(n**2),dl_list)
```

Now we have solved the problem and diagonalized the Hamiltonian. We can compare the results with the exact solution, which here can be computed straightforwardly using NumPy's eigenvalue function:

---

[3]We note that it is not so straightforward to run `Cython`-compiled code on a GPU, and as such we opted to focus on writing code which was largely CPU/GPU-agnostic using `Numba`.

[4]We use `odeint` here as it results in very simple code: for the results in the main text, we used `ode` with the `dopri5` integrator.

```
1 eig=np.sort(np.diag(sol[-1].reshape(n,n)))
2 print('Flow eigenvalues', eig)
3 print('NumPy eigenvalues', np.sort(np.linalg.eigvalsh(H0+V0)))
```

This will produce an output of the form:

```
1 Flow eigenvalues [-2.5166549  -0.77736477  1.47945136  2.30950246]
2 NumPy eigenvalues [-2.51665497 -0.77736477  1.47945126  2.30950263]
```

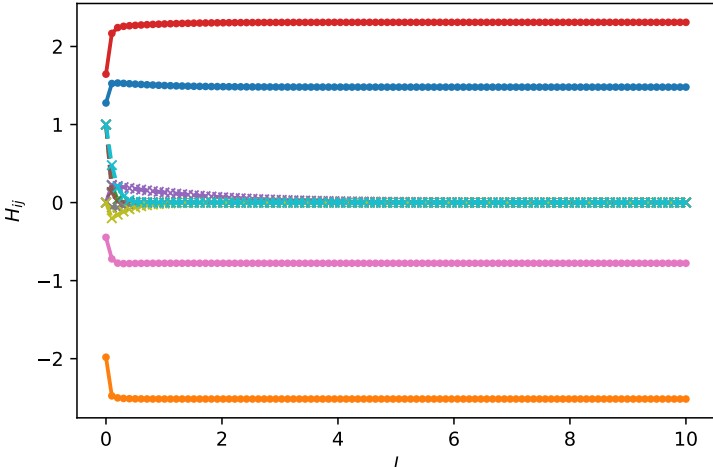

Figure 13: A sample flow of a non-interacting Hamiltonian with $L = 4$, $W/J = 2$ and $J = 1$ using the code described in Appendix. D.

Note that by the nature of (quasi)-random systems, the result will be different each time the code is run. Depending on the system size and the precise potential generated, a longer flow time may be required in order for the results to properly converge. We can also plot the flow of the Hamiltonian parameters to visualize how they behave during the diagonalization process:

```
1 sol=sol.reshape(len(sol),n,n)
2 for i in range(n):
3     # Plot the flow of the on-site terms as a solid line with
    circular markers
4     plt.plot(dl_list, sol[:, i,i],linewidth=2,marker='o',markersize
    =3)
5     for j in range(i):
6         # Plot the flow of the off-diagonal terms as a dashed line
    with cross markers
7         plt.plot(dl_list, sol[:, i,j],'--',linewidth=2,marker='x',
    markersize=5)
8 plt.xlabel(r'$l$')
9 plt.ylabel(r'$H_{ij}$')
10 plt.show()
11 plt.close()
```

This should produce an output similar to that shown in Fig. 13. We invite the interested reader to experiment with this sample code, testing different potentials and system sizes to gain a feel for how this procedure works.

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
