# Peer review of "Local Integrals of Motion in Quasiperiodic Many-Body Localized Systems"

_SciPost Physics, doi:SciPost Phys. 14, 125 (2023)_

## Round 2 · Referee Report · Anonymous (Referee 2) · 2022-11-11

Report

The authors have addressed the previous referee's points in a satisfactory way. This work is a very timely contribution to the field of MBL in quasiperiodic systems, and the results are quite interesting. The revised manuscript is also well written. I recommend publication in Scipost as is.

---

## Round 2 · Referee Report · Anonymous (Referee 1) · 2023-1-8

Report

The authors have addressed my concerns. I think that the work is a nice addition to the growing literature on the topic.

---

## Round 2 · Author Response

Warnings issued while processing user-supplied markup:

  • Inconsistency: plain/Markdown and reStructuredText syntaxes are mixed. Markdown will be used.
    Add "#coerce:reST" or "#coerce:plain" as the first line of your text to force reStructuredText or no markup.
    You may also contact the helpdesk if the formatting is incorrect and you are unable to edit your text.

Dear Editor,

Thank you for your response, and your suggestion to update the manuscript now before sending it to further referees. We have revised the manuscript to address the comments of the first referee, and look forward to receiving the reports on the updated version.

The Referee's main criticism was that it was unclear how our original manuscript differed from previous works on the topic: in the revised version, we now more clearly state that the current implementation differs not only quantitatively (faster and more accurate), but also qualitatively (inclusion of higher-order terms). This qualitative difference is precisely what allows us to investigate the phase transition by studying the form of the local integrals of motion, the main finding of the manuscript and something which was fundamentally not possible with earlier versions of this method.

We address the comments of the Referee below. We also wish to take this opportunity to say that following the first round of peer review, a potentially significant bug in our tensor contraction code was found that particularly affected the computation of the LIOMs. Out of an abundance of caution, we have replaced all of the affected data in the revised draft, resulting in cosmetic changes to several figures, most notably Fig. 7. The qualitative picture emerging from our work has not been affected and even at the quantitative level our conclusions have not been significantly altered. Due to the computational effort required, this has resulted in a lengthy delay in the submission of our revised manuscript, for which we apologise, but we believe it was necessary to ensure the accuracy of our work. For the sake of full transparency, the code has been made available as a publicly available Python package, and data used in this work has also been made publicly available (see the manuscript for details).

Yours sincerely,

S. J. Thomson and M. Schiró

Response to Referee:

We thank the Referee for their report. We are happy to see their positive comments Method explained very well" andPotentially useful method for computing LIOMs beyond 1D", as well as their feedback that the paper is interesting" andvery well written". The Referee however states that they are uncertain ``whether it reaches the acceptance criteria of SciPost Physics". In our revised manuscript, we have more clearly stated where the current method improves upon previous implementations (Refs [41,42] of the original manuscript).

The Referee's main criticisms are that it is Unclear how groundbreaking result is" and thatThe authors treat only one-dimensional models, but higher dimensions would be more interesting". We shall address the first comment below in more detail. We agree that higher dimensional systems would be interesting, but we wish to leave this for future work rather than attempt to include it in an already very long manuscript. The question of the stability of MBL in two dimensions is on much less firm ground, and we believe that this is an issue that should be studied separately rather than as a part of the current work.

The Referee summarises our work by describing it as ``new form of tensor flow equations that allows for parallelization", and while this is true, we believe that our initial manuscript did not fully convey that the possibility of parallelization is not the main step forward. In our revised version, we have attempted to make the key advantages of our new approach more clear. The central point is that by rephrasing the flow equation method in terms of tensors, this allows us to make the procedure of computing the flow more systematic than in previous works by offloading much of the algebraic complexity onto the computer. In particular, this allows us to take into account higher-order off-diagonal terms, leading to a significant increase in the accuracy of the eigenvalues obtained (by comparison with exact diagonalisation). Even more importantly, this also allows us to include higher-order terms in the LIOM operator expansion, whereas in previous works only the quadratic terms were included when computing the LIOMs. This is crucial, as it is the key step that allows us to fully incorporate many-body effects on the level of integrals of motion and estimate the MBL phase transition in terms of the structure of the LIOMs. Our previous implementations to which the Referee refers were not only less accurate that the current version, but crucially were fundamentally unable to investigate this phase transition, which is a critical question for the study of many-body localised systems and their stability in the thermodynamic limit.

Q: Compared to previous approaches with flow equations (in particular [41] by the authors), they now allow for an off-diagonal interaction term ($:c^\dagger_i c_j c^\dagger_k c_q:$) in the flow equations. In Fig. 11. they show the change with this term included, which is good. But maybe more terms are needed to get the correct LIOMs. How close are the LIOMs constructed to the real ones? It remains unclear to me. Can an estimate be made by e.g. taking the commutator with the original Hamiltonian?

A: We are not sure what the Referee means by `the real LIOMs', as the form of the LIOMs is not unique, and indeed different methods designed to compute LIOMs in different ways will in general give different results. For example, any linear combination of LIOMs will also commute with the Hamiltonian. By construction, the LIOMs obtained by this technique commute exactly with the effective Hamiltonian, and commute with the true Hamiltonian up to corrections on the order of neglected terms, i.e. $\mathcal{O}(\Delta^2)$. This can be verified numerically, however a better and more sensitive probe of the quality of the unitary transform obtained by this method is the conservation of the flow invariant, which tells us how close our approximate transformation is to a perfect unitary transform. If the transform is unitary, then the LIOMs commute exactly with the Hamiltonian. If the transform deviates from unitarity, then the LIOMs will exhibit a commensurate error.

Q: This presents potentially a very useful numerical technique, but it remains unclear how useful it is.

A: We are grateful that the Referee recognises the potential of the technique and hope that we have been able to convey a clearer picture of its use in the revised manuscript.

Q: The main advantage of the flow equation approaches over matrix product state based methods seems to be the ability to treat systems in higher than one dimension, but the authors do not do this.

A: As we mentioned above, the study of many-body localisation in higher-dimensional systems is an issue that we believe is best discussed in a separate work.

We respond to the Referee's numbered points below:

Requested changes 1- Compare and contrast to previous work better in particular [41] 2- Explain limitations of the flow ansatz -e.g. is convergence guaranteed with the ansatz given? 3- Provide clear physically highly significant (breakthrough) advantage over previous methods 4-Demonstrate error of the constructed LIOMs 5- (optional) study model beyond 1D

1 - We have more clearly stated how this work differs from previous work, in particular Ref. [41] of the original manuscript.

2 - There is a great deal of existing literature concerning this point. The Wegner generator is designed to at least block-diagonalise a Hamiltonian, as it is incapable of resolving exact degeneracies, however as these do not occur in disordered systems, in practice this choice of generator fully diagonalises our Hamiltonian. The stability of this choice of generator for disordered systems has been discussed in previous work, including our own.

3 - We believe that we have addressed this point above: the main advantage of this technique is the ability to investigate the phase transition by studying the form of the LIOMs, a crucial question in the study of localised phases which was not possible using previous approaches. The earlier flow-based methods explored by the present authors were more limited and did not allow for the higher-order terms in the LIOM operator expansion which play a key role in the present work.

4 - This is captured by the flow invariant, which parameterises how close to unitary the transform is. By construction, the LIOMs commute with the Hamiltonian.

5 - As stated above, we do not believe the study of systems beyond 1D is appropriate in the current manuscript.

Minor: a - What is the point of the normal ordering in the Hamiltonian in (1)? Can one not just write the normal ordered form to begin with?

The importance of the normal-ordering procedure is explained in Appendix A, where we point out that there is some ambiguity in the ordering of operator strings when computing commutators of high-order terms, which is resolved by adopting a normal-ordering procedure at every step of the flow. As the algebra of normal-ordered operators differs slightly from conventional fermionic operators, it is better to retain the normal-ordered form throughout. Keeping the presentation general also allows for the possibility of incorporating non-trivial normal-ordering corrections computed with respect to states other than the vacuum, something which one of us (SJT) has explored recently in arXiv:2208.11731, based upon the technique developed in this manuscript. For a comprehensive review of the algebra of normal-ordered operators and its place in continuous unitary flow methods, see S. Kehrein's book.

---

## Round 2 · List of Changes

List of changes:

i) Added paragraph to the end of the introduction to clearly state the differences between this method and previous works.
ii) Updated the lead author's affiliation.
iii) Changed the way the flow invariant is computed for Fig. 2: we now compute this fully numerically without relying on the assumption that the method has converged, giving a more accurate measure of the unitarity of the transform.
iv) Updated the data presented in Figs 1, 2, 3, 4, 5, 6, 7, 8 and 11.
v) Added a citation to the Python package PyFlow which was used to obtain the results in the current version.

---

## Editorial Decision

published